# Enhancing Antibiotic Efficacy with Natural Compounds: Synergistic Activity of Tannic Acid and Nerol with Commercial Antibiotics against Pathogenic Bacteria

**DOI:** 10.3390/plants13192717

**Published:** 2024-09-28

**Authors:** Guillermo Lorca, Diego Ballestero, Elisa Langa, María Rosa Pino-Otín

**Affiliations:** Faculty of Health Sciences, Universidad San Jorge, 50830 Villanueva de Gállego, Spain; glorca@usj.es (G.L.); dballestero@usj.es (D.B.); elanga@usj.es (E.L.)

**Keywords:** nerol, tannic acid, synergy, antibiotics, natural product

## Abstract

The search for synergies between natural products and commercial antibiotics is a promising strategy against bacterial resistance. This study determined the antimicrobial capacity of Nerol (NE) and Tannic Acid (TA) against 14 pathogenic bacteria, including ESKAPE pathogens. TA exhibited the lowest Minimum Inhibitory Concentrations (MICs) at 162.5 µg/mL against *Pasteurella aerogenes* and 187.5 µg/mL against *Acinetobacter baumannii* (WHO priority 1). NE showed its lowest MIC of 500 µg/mL against both *Pasteurella aerogenes* and *Salmonella enterica*. A total of 35 combinations of NE and 13 of TA with eight commercial antibiotics were analyzed. For NE, combinations with Streptomycin and Gentamicin were effective against *Salmonella enterica*, *Bacillus subtilis*, and *Streptococcus agalactiae*, with antibiotic MIC reductions between 75.0 and 87.5%. TA showed six synergies with Chloramphenicol, Ampicillin, Erythromycin, and Streptomycin against *Acinetobacter baumannii*, *Streptococcus agalactiae*, and *Pasteurella aerogenes*, with MIC reductions between 75.0 and 93.7%. Additionally, 31 additive effects with antibiotics for NE and 8 for TA were found. Kinetic studies on these synergies showed complete inhibition of bacterial growth, suggesting that natural products enhance antibiotics by facilitating their access to targets or preventing resistance. Given their safety profiles recognized by the EPA and FDA, these natural products could be promising candidates as antibiotic enhancers.

## 1. Introduction

Antimicrobial resistance (AMR) has emerged as one of the greatest threats to global health, economy, and development, with antibiotics losing effectiveness over the past few decades [1]. In the past two decades, antibiotic consumption has significantly increased, with a particular rise in the use of aminoglycosides. The World Health Organization (WHO) has published a comprehensive list of antibiotic-resistant bacterial pathogens [2]. This list highlights 12 families of bacteria posing the greatest threat to human health including multidrug-resistant bacteria capable of resisting three or more classes of antimicrobial drugs. This group includes bacterial species such as *Acinetobacter*, *Pseudomonas*, and several *Enterobacteriaceae* (*Klebsiella*, *Escherichia*, *Serratia*, or *Proteus*). 

The WHO has consistently urged the development of new antibiotic therapies, as the discovery rate of new antibiotics has significantly declined since 1960 [3], despite efforts from the pharmaceutical industry. One viable strategy to address this challenge is the exploration of natural products (NP) from plants, which are, historically, a rich source of active ingredients [4,5] and traditional phytotherapies [6]. Each plant species can produce between 500 and 800 different secondary metabolites [7,8], many of which have known antimicrobial properties [9,10]. Some commercial antibiotics exhibit synergistic effects when combined with NPs [11,12,13], resulting in enhanced efficacy compared to single agents [14,15,16]. Such combinations could supplement traditional antibiotic treatments [17,18] and reduce the environmental impact of antibiotics reaching aquatic and terrestrial environments [19,20], as well as the spread of resistance genes. Furthermore, plant-based NPs align with the “One Health” concept, causing fewer side effects on human, animal, and environmental health [21,22].

Within the realm of plant secondary metabolites, many NPs derived from essential oils have demonstrated antifungal and antimicrobial properties [23,24]. These compounds, often extracted via hydrodistillation [25,26], include phenolic compounds, diterpenes, flavonoids, and volatile terpenes. While essential oils concentrate low-water-solubility NPs, some effective antimicrobial compounds are found in the aqueous by-products or hydrolates [27,28,29,30].

Nerol (NE), a volatile monoterpene (Z)-3,7-dimethylocta-2,6-dien-1-ol), is found in plants like lemongrass and hops and is widely used in food, cosmetics, and household products due to its Generally Recognized as Safe (GRAS) status by the FDA [31]. NE exhibits broad-spectrum antimicrobial activity against various Gram-positive and Gram-negative bacteria [32,33,34], as well as antifungal properties against *Candida albicans*, *Aspergillus* species, and others [34,35,36,37,38]. NE is a valuable ingredient in fragrances [31], cosmetics, soaps, and shampoos, and it is also present in cleaning products [39]. Notably, NE has shown synergistic potential with norfloxacin, significantly reducing its MIC against *S. Aureus* (*Staphylococcus aureus)* [40], and has been used in synergy with other natural products such as carvacrol against nosocomial pathogens [33,34].

Tannic Acid (TA), a naturally occurring polyphenol, is extracted from various plants used as food and feed [41,42]. Recognized for its antioxidant, antimutagenic, and antitumoral properties [43], TA also displays broad-spectrum antimicrobial activity against numerous bacteria and fungi [42,44]. Additionally, it has recently received attention due to its intrinsic properties such as polymerization, antioxidation, and metal chelation in applications of engineered advanced materials in biomedicine [41]. It is also marketed in pharmaceutical products for the treatment of diarrhea, such as Cesinex^®^, demonstrating a broad spectrum and a good safety profile [45]. Although the literature mainly focuses on synergistic effects with plant extracts containing this product as one of many constituents, there is some evidence in the literature suggesting a potential synergistic activity of TA with antibiotics against a few bacteria such as *P. aeruginosa (Pseudomona aeruginosa)* or *S. aureus* [44,46,47].

Therefore, these two natural products are good candidates in the strategy to reduce antibiotic consumption through synergistic combinations. However, studies on their antimicrobial capacity in combination with commercial antibiotics are scarce. They have only been conducted on a few bacteria and are rarely characterized beyond the calculation of Minimum Inhibitory Concentration (MIC) differences.

Hence, the aim of this study is to identify and characterize for the first time the potential synergies of these two natural products with a broad spectrum of commonly used antibiotics and pathogenic bacteria. To this end, (1) the antimicrobial effect (bactericidal and bacteriostatic) of NE and TA has been quantified on 14 reference bacterial strains responsible for the most common infections today [48], which have been described as having a high potential to develop antimicrobial resistance, according to the WHO’s list of priority pathogens [2]; (2) the synergistic effects with eight commonly commercialized antibiotics (ABXs), which represent various modes of action, have been analyzed on these 14 bacteria; and (3) the kinetics of inhibition of the most promising combinations have been analyzed.

## 2. Results

### 2.1. Antimicrobial Activity of Tannic Acid, Nerol, and Antibiotics

The antibacterial activity measured as Minimum Inhibitory Concentration (MIC) of Nerol (NE) and Tannic Acid (TA) against 14 microorganisms is shown in Table 1. A previous toxicity test assessment of dimethyl sulfoxide (DMSO) showed that the growth of none of the tested bacteria, except for *Proteus mirabilis* (and thus ruled out for the synergy tests) was affected by a concentration of 2.5% DMSO used to solubilize natural products (NPs).

Both NPs alone showed antimicrobial activity against most of the tested bacteria. TA had the greatest effect on the bacteria studied, presenting MICs with a value ≤ 500 μg/mL in four of the bacteria studied, where the lowest value was 162.50 μg/mL for *Pasteurella aerogenes* and 187.5 μg/mL for *Acinetobacter baumannii*, classified as priority 1 by WHO. NE had MICs of 500 μg/mL in two cases, *Salmonella enterica* and *Pasteurella aerogenes.*


The values of the ratio between the Minimum Bactericidal Concentration (MBC) and the MIC of NE and TA indicated that the activity was bactericidal in most cases (MBC/MIC ≤ 4) for both compounds.

Appendix A shows MIC test results for the 8 commercial antibiotics (ABX) against 13 different pathogenic bacteria. These concentrations will be used to calculate the fractional inhibitory concentration index (FIC_I_) for the combinations with synergistic effects.

### 2.2. Checkboard Tests for Synergy Assessments between Tannic Acid, Nerol, and Antibiotics

Potential reductions in MIC for commercial ABXs in the presence of NE or TA were evaluated by calculating the FIC for each combination, as shown in Table 2 and Table 3. Although most interactions with NE were found to be additive, four combinations exhibited synergistic effects: *S. enterica (Salmonella enterica)* + Streptomycin (STM), *B. subtilis (Bacillus subtilis)* + Gentamicin (GTM), and both combinations for *S. agalactiae (Streptococcus agalactiae)* (with STM and GTM). FIC values were a maximum of 0.5, with ABX MIC reductions of 87.50% for *B. subtilis* and a reduction of 75.00% in all other cases. The NE reductions were 75.00% in all cases except for *B. subtilis*, which had a reduction of 87.50%.

Table 3 shows that six additional synergistic interactions were found for TA. Chloramphenicol (CHL), Ampicillin (AMP), and Erythromycin (ERY) exhibited synergism with *A.baumannii (Acinetobacter baumannii)*, while STM and CHL exhibited synergism with *S. agalactiae*, and CHL with *P. aeruginosa*. The nature of the remaining eight combinations tested was additive. The combination of TA and AMP with *A. baumannii* achieved the lowest FIC value (0.312). ABX MIC reductions ranged from 75.00% to 93.75%, while TA reductions ranged from 75.00% to 87.50%. No antagonistic interactions were observed in the results. 

From the synergy checkerboard results, isobolograms have been plotted for TA and NE (Figure 1 and Figure 2) in cases where the FIC_I_ ≤ 0.5. Four combinations show two synergy points, *A. baumanii* + TA + CHL, *A. baumanii* + TA + ERY, *S. agalactiae* + TA + CHL, and *B. subtilis* + NE + GTM. The lowest FIC_I_ was 0.250 for the combination *A. baumanii* + TA + CHL.

### 2.3. Synergy Kinetics Study

A kinetic study of the 10 different synergistic combinations was carried out, with the aim of gaining a deeper understanding of the behavior of each bacterium throughout its incubation period. Results are shown in Figure 3 and Figure 4.

## 3. Discussion

In this study, we investigated the ability of two plant-derived natural products to synergize with widely used commercial ABXs, aiming to reduce the required ABX dosage while maintaining efficacy. TA and NE were selected based on their demonstrated antimicrobial activity against various Gram-positive and Gram-negative pathogens [31,49,50]. The previous literature has also begun to suggest potential synergistic effects of these compounds with one or a few ABXs [51]. The use of these products offers a range of advantages: NE is abundant in essential oils from widely cultivated plants [52], while TA is prevalent in the bark of trees like oak and chestnut mahogany [53]. Therefore, their extraction from these inexpensive plant materials ensures high availability. Moreover, the essential oil industry enables cost-effective NE production, often as a byproduct of other compounds. Similarly, byproducts from the wine and wood industries can be used to extract TA, promoting sustainable production [54].

The antimicrobial capacity of both products was tested against a comprehensive and representative panel of 14 pathogenic bacteria to determine the MICs. This allowed us to conduct a synergy study with 8 ABXs, resulting in a total of 48 combinations. The synergies (4 for NE and 6 for TA) were further analyzed by monitoring their growth kinetics. A reference bacterial strain was selected for each bacterium studied, enabling comparison with results from the literature, such as those involving other natural products or combinations with different ABXs.

### 3.1. Antimicrobial Activity of Tannic Acid and Nerol

NE and TA exhibit antimicrobial activity against most of the tested bacteria and the fungus *Candida albicans*: NE showed activity against 11 out of 14 organisms, with MIC ranges from 500 to 2000 µg/mL, and TA showed activity against 6 out of 14 organisms, with MIC ranges from 325 to 1800 µg/mL (Table 1).

The results obtained from MICs have been compared with the literature; however, the existing literature is scarce, as it mainly focuses on essential oils rather than pure compounds. The value of 500 μg/mL has been considered a reference value to consider a strong effect of NP according to bibliographic criteria [55].

The MIC values obtained for NE against *S. enterica* and *P. aerogenes* (500 μg/mL) are very similar to those found in the literature (441 μg/mL and 600 μg/mL) [32,56]. Similarly, the effect of NE on *E. faecalis(Enterococcus faecalis)*, *P. aeruginosa*, *S. enterica*, *K. pneumoniae (Klebsiella pneumoniae)*, and *C. albicans* has been documented in the literature, showing very similar MICs (600 μg/mL vs. 1000 μg/mL) [56].

Regarding TA, the literature shows a MIC for *S. aureus* between 40 and 160 μg/mL [44], which differs significantly from our results (325 μg/mL). There are also differences compared to *A. baumannii*, where our results show 187.5 μg/mL, while the literature reports a MIC of 600 μg/mL [57]. The existing variability may be due to the technique used, the solvent, the strain, or the culture medium [58].

To our knowledge, there are no bibliographic MIC values for the other bacteria studied in our work. There are studies that use essential oils containing these products, but these results are not comparable, as essential oils contain other compounds that can affect the calculation of MICs. In the case of *P. mirabilis (Proteus mirabilis)*, we were unable to calculate MICs because the concentration of DMSO needed to dissolve our NPs affected the bacteria.

Considering these results, two inclusion criteria were established to test synergies. 

ABXs with a MIC below 4 µg/mL for certain bacteria were discarded, mainly due to the difficulty in assessing reductions in such low concentrations;Bacteria susceptible to DMSO at concentrations ≥2.5% were discarded, as it was not possible to have stable dilutions of either NE or TA.

Once these criteria were applied, 35 interactions with NE and 13 with TA were selected for testing in checkerboard assays (Table 2 and Table 3). The synergies identified between natural products and antibiotics have been represented in isobolograms, which allow for a more intuitive visualization of the relationship between individual data points and a reference line (the addition line), making it easier to understand the degree of synergy for dose combinations that fall below this line.

### 3.2. Kinetic Study of the Obtained Synergies

In all cases, as expected, no bacterial growth was observed for the MICs of ABXs and NP alone (orange and light blue lines in Figure 3 and Figure 4). Additionally, while the ABX and NP separately produce a significant antimicrobial effect at the synergistic concentration compared to the control (red and purple lines in Figure 3 and Figure 4), their combination at the same concentrations (green line) achieves complete inhibition of the growth of all tested bacteria. Consequently, these last three growth curves, which represent nearly complete inhibition of microbial growth, often overlap. Although natural products have shown clear antimicrobial activity on their own, the required concentrations are high. Combining them with ABXs not only reduces the MIC of the ABX, which is our primary goal, but also lowers the necessary concentrations of the natural product while maintaining the same antimicrobial effectiveness. This allows for a synergistic combination with low doses of both agents. Additionally, the individual curves of NP and ABX indicate the potential mechanisms of action of each product separately, as explained below.

### 3.3. Tannic Acid and Nerol Synergies with Antibiotics against Gram-Positive Bacteria 

The antimicrobial capacity of TA has been explained on the basis of its ability to damage the cell membrane, causing lysis of the bacteria, inhibiting enzymes of bacterial metabolism, affecting protein synthesis and bacterial growth [59], disrupting oxidative phosphorylation in mitochondria [60], chelating metal ions necessary for bacterial growth [17], and inactivating bacterial adhesins, which are responsible for the adherence of bacteria to the host [61,62]. Other suggested impacts of TA on microorganisms include the inhibition of efflux pumps, as observed in the case of *S. aureus* [46]. In addition to all these mechanisms, the literature shows a dependence on the concentration present, the pH, and the matrix in which it is found [63]. Recent studies on the genomics and transcriptomics of the mechanism of action of Tannic Acid and other Gram-positive cocci, such as *S. aureus* [64], have revealed that the integrity of the cell envelope is affected by a decrease in the expression and protein abundance of enzymes involved in the synthesis of peptidoglycans, teichoic acids, and fatty acids. Additionally, there is a reduction in ribosomal components that impacts protein synthesis.

The mechanism of action of NE is also based on interaction with the cell membrane. Similar to other acyclic monoterpenes, it is capable of interfering with the integrity of the membrane, leading to the development of pores and even destruction, resulting in leakage of cell contents [65], which has been primarily studied in fungi [34,38]. NE has been shown to form aggregates of antimicrobial–lipid complexes, reducing lipid packing efficiency, increasing membrane fluidity, and altering the total dipole moment of the membrane [66]. Its lipophilicity also enables it to partition into the lipophilic lipids of the mitochondria, disturbing these structures.

Gram-positive bacteria are affected by TA, as demonstrated in our results for *S. agalactiae* (Figure 3a,b). TA has been reported to influence the growth of other Gram-positive bacteria such as *Listeria monocytogenes* [42] and *S. aureus* [67,68], where it can also inhibit biofilm formation [69]. NE has also shown its antimicrobial capacity against Gram-positive bacteria, including *S. aureus* [32] and *S. epidermidis (Staphylococcus epidermidis)* [39].

In Figure 3a,b, TA at a synergistic concentration, while having minimal impact on total growth (Cmax similar to control), can delay growth by a few hours (r_TA_ = 0.676 compared to r_control_ = 1.040, in Figure 3a, for example), likely due to cell envelope damage that the bacteria eventually recover from. For NE with *S. agalactiae* (Figure 4c,d), a similar but less pronounced effect is observed.

The ABXs (STM, CHL, and GTM) at this sub-MIC synergistic concentration show little impact on microbial growth. Only CHL (Figure 3b) significantly reduces the total population growth (Cmax_CHL_ = 1.770 compared to Cmax_control_ = 2.191). These ABXs target bacterial ribosomes. CHL is a broad-spectrum ABX that inhibits protein synthesis by binding to the 50S subunit of bacterial ribosomes, preventing peptide bond formation. Streptomycin (STM) and Gentamicin (GTM) both target the 30S subunit, disrupting protein synthesis by causing mRNA misreading and incorporating incorrect amino acids, leading to defective proteins and bacterial cell death [70]. Both ABXs are more effective against Gram-negative bacteria, and at this sub-MIC concentration, they possibly do not reach their target, which together explains their minimal effect on bacterial growth that we observed.

However, in synergy with TA and NE, the effect of these ABXs is enhanced to the point of completely inhibiting bacterial growth (green line in Figure 3 and Figure 4). The most plausible mechanism for this synergy could be that the membrane-disrupting activities of both TA and NE facilitate the ABXs’ access to their intracellular ribosomal targets, making the ABX effective even at sub-MIC concentrations. Additionally, TA can inhibit genes that regulate the synthesis of proteins composing the *S. aureus* 50S and 30S ribosome, as well as genes for proteins such as the translation initiation factor IF-2, which is involved in regulating the efficiency and fidelity of translation–initiation complex formation [64]. On its own, TA would only cause the growth delay we observed, but in combination with the ABX, the effects of both products on the ribosomal target would accelerate the impact on protein synthesis, reducing bacterial survival. Furthermore, it has been reported that TA could enhance the inhibitory effect of ABXs targeting ribosomal sites, such as Erythromycin, on drug-resistant *S. aureus* by inhibiting bacterial virulence factors like efflux pumps [46], which would promote ABX accumulation in the cytoplasm, making it effective even at sub-MIC concentrations.

In the case of *B. subtilis* exposed to sub-MIC concentrations of NE (Figure 4b), we observe minimal changes compared to the control. To our knowledge, there are no previous studies specifically examining the activity of NE against *B. subtilis*, except for essential oils containing fractions of this product, which have demonstrated slight antimicrobial activity against *B. subtilis* [71]. When exposed to sub-MIC concentrations of GTM, however, we observe that the ABX can delay growth for several hours, but the bacteria eventually recover, reaching a Cmax that is like or even higher than the control (Figure 4b). This recovery might be due to enzymatic modification of the ABX (aminoglycoside-modifying enzymes), alteration of the target site, or active efflux of the ABX, as described in other cases for this ABX in aminoglycosides [72]. It is possible that after prolonged exposure to the ABX, resistant strains emerge through competitive selection. However, this strain typically does not have intrinsic resistance to Gentamicin (https://genomes.atcc.org; 8 July 2024), and horizontal gene transfer is unlikely in a reference strain, necessitating further studies to clarify the mechanisms of recovery at sub-MIC concentrations. In synergy, however, NE can eliminate this effect by enhancing the ABX’s activity, possibly by facilitating its access to the ribosomal target, or by inhibiting bacterial resistance mechanisms.

### 3.4. Tannic Acid and Nerol Synergies with Antibiotics against Gram-Negative Bacteria 

Gram-negative bacteria have a more complex envelope, with an external lipid bilayer and a peptidoglycan layer [73,74,75]. It would be expected that they are somewhat more resistant to the membrane disruption effects of TA. However, TA appears to be much more effective against *A. baumannii* and *P. aerogenes (Pasteurella aerogenes)* (Figure 3c,d,f) than we observed with Gram-positive bacteria. This is consistent with the literature describing TA’s effects on various Gram-negative bacteria, such as *E. coli (Escherichia coli)* [51] and *S. enterica* [76].

Despite its high molecular weight (1701.2 g/mol) [77] and high hydrophilicity (log Kow = −0.19) [77], TA contains multiple hydroxyl groups that can electrostatically interact with the phosphate and carboxylate groups present in the LPS of the Gram-negative outer membrane. This interaction could facilitate the binding of TA to the bacterial surface, affecting the permeability and integrity of the outer membrane, allowing it to penetrate through the peptidoglycan layer to reach the cell membrane and its cytoplasmic targets. Interestingly, quorum-sensing disruption effects might also occur, as observed with TA in a P. aeruginosa strain [78].

In the case of *A. baumannii* (Figure 3c,d), the effect of sub-MIC concentrations of TA appears more detrimental to the bacterial population from the onset of bacterial growth following the lag phase, resulting in a lower Cmax_TA_ compared to the control in all three cases. Additionally, it is evident that both AMP and CHL inhibit bacterial growth for 3 to 4 h longer than TA, reflected in a higher r. However, the microbial population exposed to ABX is able to recover better, with Cmax values closer to the control.

CHL targets the 50S subunit of the bacterial ribosome, while AMP is a broad-spectrum penicillin ABX that disrupts bacterial cell wall synthesis by binding to bacterial penicillin-binding proteins (PBPs), preventing cross-linking of peptidoglycan chains, and leading to bacterial lysis [79]. It seems that the bacteria can recover from ABX exposure after 5 to 10 h, possibly due to resistance mechanisms. Indeed, *A. baumannii* possesses various intrinsic resistance genes against all three ABXs. For instance, it has inactivating enzymes against CHL (chloramphenicol acetyltransferase) and efflux pumps that expel CHL and ERY from the cell, as well as beta-lactamases for AMP, among others [80]. 

We observe that the growth of *A. baumannii* is halted after exposure to TA from 10–15 h onward, indicating that the effect of this natural product remains consistent over time, possibly due to its lower potential to induce resistance, as seen with other similar natural products [24].

The effect of ERY on the same bacteria (Figure 3e) shows less growth delay compared to the previous cases, despite the ABX mechanism of action being similar to that of CHL. ERY is a broad-spectrum ABX of the macrolide family, which acts by binding to the 50S ribosomal subunit of sensitive microorganisms, similar to CHL, but at a different site, inhibiting ribosomal translocation and thus the incorporation of new amino acids, ultimately resulting in the arrest of peptide chain elongation.

When *P. aerogenes* is exposed to TA and CHL (Figure 3f), we observe that TA not only delays microbial growth more than the ABX (r_TA_ = 0.187 vs. r_CHL_ = 0.423), but also results in a more pronounced inhibition of the total microbial population (C_maxTA_ = 0.826 vs. C_maxCHL_ = 1.834) compared to the previous cases. It is important to note, however, that in both cases, the dose of TA in synergy is somewhat higher than that of the ABX. The action of the ABX on this bacterium is also somewhat less effective.

As seen in the case of Gram-positive bacteria, the combination of the ABX and the natural product leads to complete inhibition of bacterial growth (green line in Figure 3 and Figure 4). Combinations of TA likely enhance or facilitate the interaction of an antimicrobial agent with its target inside the pathogen, thereby preventing the emergence of resistance. For CHL and ERY, TA seemingly facilitates their access to their intracellular ribosomal targets by damaging the cell membrane. In the case of AMP, its mechanism of inhibiting peptidoglycan wall synthesis prevents the bacteria from maintaining cell wall integrity, leading to unbalanced osmotic pressure within the cell, ultimately causing cell lysis, which is probably further accentuated by TA’s membrane-altering effects.

In the case of NE, its antimicrobial activity against Gram-negative bacteria such as *E. coli* and other intestinal bacteria [49], as well as *S. enterica* [32], has been reported. However, at the synergistic concentrations used in our study, we detected very little effect of NE on *S. enterica*, with growth kinetics like the control (Figure 5a). The properties of NE, such as its high lipophilicity (log Kow = 3.47) [77] and its low molecular weight (154.25 g/mol), facilitate its interaction with cell membranes. Gram-negative bacteria have porins in their outer membrane that act as selective channels, allowing the passage of certain small hydrophilic solutes. NE, being lipophilic, may have difficulty passing through these porins, although its small size might help. The outer membrane of Gram-negative bacteria and the presence of liposaccharids can present an additional barrier that may reduce the efficiency with which NE traverses these structures.

However, the antimicrobial effect of STM is very pronounced, both in reducing the time to the onset of growth by 12–13 h and in decreasing C_max_ by more than 50% compared to the control, probably because NE’s membrane-altering activity facilitates the access of STM to its 30S ribosomal target in the bacteria. In addition, genes such as aadA1, aadA2, and strA are associated with STM resistance in various *Salmonella* strains [81] as well as in ours (https://genomes.atcc.org). These genes typically encode enzymes that modify and inactivate streptomycin, preventing it from binding effectively to the bacterial ribosome.

Finally, it is worth noting that although our study has focused on synergies, given that additive interactions may not be as effective as synergistic interactions, the latter are far more numerous in our results and therefore deserve attention.

As seen in Table 2 and Table 3, they also allow for a reduction in the concentration of ABX, and in many cases, the concentration of the adjuvant needed to achieve additivity in combinatory treatment might even be lower than what is observed in synergistic interactions. 

Although the mechanisms of additive activities have been little studied, we hypothesize that one possible cause could be that the action of the natural product only slightly damages the membrane or causes some intracellular damage, which by itself does not facilitate the ABX’s action but simply adds to the damage inflicted on the bacteria. Additionally, due to the broad, nonspecific mechanisms associated with natural products, there may not be an opportunity for the combined activities of these compounds to exceed the sum of their parts, as has been suggested in the case of disinfectant combinations [82]. However, another study [83] argues that the effects on the membrane from the additive interaction of cinnamon bark oil and meropenem are very similar to those observed in previously reported synergistic combinations, indicating that further studies are indeed necessary to clarify this point. It is important to assess the therapeutic potential of additive interactions alongside synergistic ones, as many studies on natural ABX adjuvants, including this one, report an equal or greater number of additive interactions [83,84,85,86,87,88]. 

### 3.5. Future Perspectives

These results propose both TA and NE as enhancers of the activity of commercial ABXs as well as Antimicrobial Resistance Modifying Agents.

In addition to the previously mentioned advantages in their production, TA and NE likely have a lower potential for inducing resistance compared to commercial ABXs as a result of several factors [89]. First, natural compounds often have more complex and diverse chemical structures than synthetic ABXs, making it harder for bacteria to develop effective resistance mechanisms, as they would need to adapt to multiple sites or modes of action, which is more challenging. Additionally, while synthetic ABXs typically target a single cellular process, natural products like TA and NE act on multiple fronts, as we have seen. This multifaceted approach further reduces the likelihood of resistance, as bacteria would need to simultaneously mutate in several areas. Moreover, NE and TA can disrupt bacterial membranes, affecting resistance mechanisms associated with these membranes, such as efflux pumps. Another key point is that natural compounds have been in contact with microorganisms for millennia, possibly leading to an evolutionary balance where bacteria are less prone to develop resistance. In contrast, synthetic ABXs are more recent and often used in large quantities, which can exert intense selective pressure, quickly fostering the development of resistant strains. When natural products are used in synergy with ABXs, the required ABX doses can be significantly reduced, as seen in this study, decreasing the chances of resistance development. Finally, the simultaneous action of natural products and ABXs on different cellular targets makes it more difficult for bacteria to develop resistance strategies against both, a benefit that is less common in synthetic ABXs typically used as monotherapies.

Although further studies are necessary to clarify the mechanisms of action of these products in synergy with each type of ABX, the fact that both products can be considered safe for human use and are already marketed as health products makes them very promising candidates for clinical application. TA is already marketed as a medical product [90] and is recognized as safe in the US by the FDA (Food and Drug Administration) [91] and in the European Union by the EFSA (European Food Safety Authority) [92]. On the other hand, NE was approved by the Food and Drug Administration as a flavor (21 CFR 172.515) and is recognized as a safe flavor ingredient—GRAS 3 (2770)—by the Flavor and Extract Manufacturers Association (FEMA, https://www.femaflavor.org/flavor-library; 6 June 2024). Although it is not specifically approved as a food additive by regulatory agencies such as the FDA or the EFSA, the recommended doses for its use are much higher than the concentrations we have obtained in our synergies [31].

In general, natural products require higher doses compared to ABXs to achieve antimicrobial activity. However, the advantage of synergistic combinations is that they can reduce not only the concentration of ABXs but also the amount of the natural product needed, as demonstrated by our results.

Typically, these synergies are considered for use in topical formulations such as lotions, ointments, gels, or creams for skin infections, wounds, and ulcers [89]. They could also be applied in mouthwashes and oral rinses [93]. Oral solutions like Cesinex^®^ [45] are already marketed for gastrointestinal conditions, with Tannic Acid as one of their components.

Additionally, there is potential for these combinations in veterinary feed to treat animal diseases, which is relevant given EU Regulation 2019/6 [94], which emphasizes reducing ABX use in livestock to mitigate environmental impact [20] and microbial resistance [95]. Furthermore, these synergies could be valuable as disinfectants to inhibit microbial biofilm formation on stainless steel surfaces [96].

The effective doses found in the synergies between natural products and ABXs are very low, ranging from 23.44 to 81.25 μg/mL for TA and from 125 to 250 μg/mL for NE. The ABXs in synergy further reduce their concentrations significantly, with ranges between 0.78 and 31.25 μg/mL. Topical ABXs available on the market, for example, are used at concentrations several orders of magnitude higher than the synergistic concentrations obtained in this study. Most of the available ABX ointments (such as those with mupirocin or sodium fusidate) and antifungal ointments (like ketoconazole) contain a concentration of 20 mg/g. Other formulations contain even more, such as oxytetracycline or clindamycin ointments, which typically have 30 mg/g of the ABX. Commercially available ABX eye drops have similar concentrations, often 3 mg/mL (or 3000 µg/mL). Some formulations of Gentamicin even reach up to 5000 µg/mL of the ABX. Therefore, it is pharmacologically realistic to prepare topical, veterinary, or even oral formulations containing these ABXs and natural products at the effective concentrations found in the synergies of our study.

New perspectives on the antimicrobial use of these natural products are emerging. TA, a type of polyphenol, can form metal complexes through coordination, as has been observed with other polyphenols such as gallic acid [97,98]. TA can form coordination compounds with silver and iron, leading to nanoparticles with antimicrobial activity [99,100]. This opens the door to potential new antimicrobial applications for these natural products, such as in wound management [101] or in shoe insoles and fruit preservation [100]. Additionally, the efficacy of these nanoparticles could potentially be enhanced by combining the natural product with an ABX to achieve synergistic effects.

However, before developing clinical formulations, it is crucial to assess the potential toxicity of the combination, as well as its bioavailability, through clinical trials [102], since the physicochemical and pharmacological properties of these combinations may differ from those of the individual products [103]. Formulations must also take into account organoleptic properties and stability. Finally, within a One Health framework, it is important to evaluate whether these synergies are more environmentally friendly compared to using higher doses of ABXs alone.

## 4. Materials and Methods

### 4.1. Antimicrobial Compounds

Two NPs, NE and TA, and eight ABXs were selected as antimicrobials for this study. Figure 5 shows the chemical structures of both NPs. CAS number, provider, and some chemical properties of each antimicrobial are compiled in Table 4. DMSO (Fisher Bioreagents) has been used to solubilize the NP solutions. The natural product solutions have been prepared with a concentration of 5% DMSO.

### 4.2. Microorganisms

Fourteen reference bacterial strains, all well-known human and veterinary pathogens, were chosen for this study. Selected strains included both Gram-positive (*Staphylococcus aureus* ATCC 9144, *Streptococcus agalactiae* ATCC 12386, *Listeria monocytogenes* ATCC 7644, *Enterococcus faecalis* ATCC 19433, and *Bacillus subtilis* ATCC 6633) and Gram-negative (*Escherichia coli* ATCC 25922, *Salmonella typhimurium* ATCC13311, *Pseudomona aeruginosa* ATCC 27853, *Acinetobacter baumannii* ATCC 19606, *Klebsiella aerogenes* ATCC 13048, *Pasteurella aerogenes* ATCC 27883, *Klebsiella pneumoniae C6*, *Proteus mirabilis*, ATCC 35659, and *Serratia marcescens* ATCC 13880).

All microorganisms were purchased as freeze-dried Culti-LoopsTM bacteria from Thermo Scientific (Dartford, UK), rehydrated, and kept at −80 °C in cryovials (Deltalab S.L. Barcelona, Spain) until used. Thermo Scientific and ATCC product sheet instructions for each strain were followed for rehydration process and culture conditions, as well as antibacterial activity testing (www.atcc.org, accessed on 20 May 2024).

### 4.3. Determination of the Antimicrobial Activity: Minimum Inhibitory Concentration (MIC) and Minimum Bactericidal Concentration (MBC)

To study antimicrobial properties, Minimum Inhibitory Concentrations (MICs) were determined using the broth microdilution method in 96-well round-bottom microplates (Deltalab S.L. Barcelona, Spain), according to the ISO 207776-1 [104] and the Clinical and Laboratory Standards Institute’s [105] (CLSI, M100-S15 2018) guidelines. Each step of the procedure was performed in a flow chamber (Model MSC Advantage 1.2) under sterile conditions. The MBC (Minimum Bactericidal Concentration) index, which indicates the minimum concentration capable not only of stopping bacterial growth, but also of eliminating bacteria, was calculated by culturing the contents of several wells of MIC plates in Petri dishes for 24 h.

ABX stock solutions were prepared in distilled water (SIEMENS Ultra Clear™), adding 5% of DMSO (CAS: 67-68-5) from Fisher Bioreagents (Madrid, Spain), with a purity of 99.7% for the stock solutions of NE and TA. A toxicity assessment of DMSO was previously tested on every bacterial strain, assessing that its highest concentration in the wells (2.5%) did not affect microbial growth [12].

Using a BioTekTM Synergy H1 hybrid multimode microplate reader (625 nm), bacterial cultures were previously adjusted to the McFarland standard (CLSI, 2018) [105] to achieve a standard initial bacterial concentration per well of roughly 2.5 × 10^5^ CFU/mL. 

Microplates were cultured for 24 h at the appropriate temperature for each bacterium in a bacteriological culture incubator (Incuterm, Trade Raypa^®^). According to CLSI guideline M07-A9 (2018), the MIC was defined as the lowest concentration that visibly inhibited microbial growth. The absorbance of each well was also measured at 625 nm using a microplate reader to provide a more precise measurement of microbial growth. 

MBC/MIC ratio indicates the bacteriostatic or bactericidal activity of a compound on each bacterium, i.e., whether it causes the death of microorganisms or only inhibits their growth. If this ratio is ≤4, a substance is considered to have bactericidal activity [106,107].

### 4.4. Determination of the Natural Product-ABX Combination Behavior

#### 4.4.1. Checkboard Assays and Fractional Inhibitory Concentration Index

Synergies to be tested were selected based on MIC experiment results for ABXs and NPs, choosing those combinations with a suitable solubility for the NP. To measure these potential synergies between ABXs (drug A) and the NP (drug B), the checkerboard method was chosen. NP (NE or TA) were diluted vertically, from columns 1 to 7 of the 96-well plate. The matching ABXs for that synergy were then diluted vertically from rows A to G. Stock concentrations prepared for every substance were four times higher than its MIC for each bacterium tested, to ensure a reliable synergy value.

The FIC_I_ index, which defines the type of interaction produced between two drugs, was calculated as follows:(1)$$FIC_{I}=FIC_{A}+FIC_{B}=\frac{MIC_{A+B}}{MIC_{A}}+\frac{MIC_{B+A}}{MIC_{B}}$$

In this equation, drug A is a NP (NE or TA), and B is a commercial ABX. Then, FIC_A_ is the quotient between the MIC of drug A in the presence of drug B (MIC_A+B_) and the MIC of drug A alone (MIC_A_). FIC_B_, on the other hand, is the quotient between the MIC of drug B in the presence of drug A (MIC_B+A_) and the MIC of drug B alone (MIC_B_).

Following the guidelines from the European Committee on Antimicrobial Susceptibility Testing [108], a synergy is defined as an interaction with a FIC_I_ value of ≤0.5. Values between 0.5 and 1 correspond to additivity, values ranging from >1 to 2 indicate no significant interactions, and values ≥2 indicate antagonistic effects [109,110].

#### 4.4.2. Growth Kinetic Tests

To provide a more accurate evaluation of the effects of synergistic combinations (those with a FICI ≤ 0.5), growth kinetics tests were conducted. As described in Section 4.3, bacterial cultures were adjusted to the McFarland standard. Then, based on the data provided by the checkerboard test, new 96-well microplates were prepared, exposing bacteria to various inhibitory and sublethal concentrations of ABXs, NPs, and their combinations [93]. Afterward, the plates were then introduced to the absorbance reader Synergy H1 (Biotek), which incubated them for 24 h at specified temperatures and simultaneously took one absorbance reading every hour. Data were plotted as absorbance vs. time, obtaining the growth/time curves. Each absorbance reading consisted of four replicates. Kinetic curves were fitted to a logistic model (Equation (2)) for sigmoid microbial growth: (2)$$Absorbance=\frac{C_{max}}{1+e^{b-rt}}$$
where C_max_ is the maximum population density achievable, r is the rate of population increase, and b is the fitting parameter. C_max_ and r were calculated to determine the kinetics growth of synergistic combinations.

The significance of Kinetic curve differences compared to the control was assessed using ANOVA for parametric data, conducted with SPSS software (version 28.0.1.0, 142).

#### 4.4.3. Isobolograms

An isobologram is a graphical representation that allows the observation of ABX-natural product interactions from the results obtained in the checkerboard tests used to obtain the synergies [111,112]. In this work, only isobolograms whose FICI is ≤0.5 are plotted.

To draw an isobologram, the MIC of the NP is placed on the X-axis and the MIC of the ABX on the y-axis. The graph is plotted with the combinations obtained in the checkerboard tests that inhibit bacterial growth. It includes an ‘addition line’ (solid line) that helps differentiate between additive effects (where points fall above or near this line), synergistic effects (indicated by concave isoboles below the line), and antagonistic effects (shown by convex isoboles above the line). Additionally, there is a lower dotted line that marks the boundary of synergy. Points situated above or below this dotted line represent different degrees of synergistic interaction.

## 5. Conclusions

Although natural products have repeatedly demonstrated antimicrobial capabilities in the literature, the fact is that they require high doses that hinder clinical application, making it difficult for them to replace much more efficient commercial ABXs. However, the strategy of combining natural products with commercial ABXs may be a good solution to combat antimicrobial resistance. Our results show that both TA and NE, in combination with widely used commercial ABXs, are capable of completely inhibiting microbial growth, reducing the ABX dose by margins from 75.00% to 93.75%. Additionally, the dose of the natural product is also reduced by about 75% in the synergies. The growth kinetics of the microbes when treated with the two products separately and in synergy suggest different mechanisms of action, some of which indicate the natural product’s ability to enhance the ABX’s activity by making the target more accessible or even acting as Antimicrobial Resistance Modifying Agents.

The fact that both natural products are considered safe by international official agencies and that their doses are significantly reduced in synergy, thereby reducing their toxicity, makes their application as enhancers of commercial ABXs very promising for clinical, food, or veterinary use.

## Figures and Tables

**Figure 1 plants-13-02717-f001:**
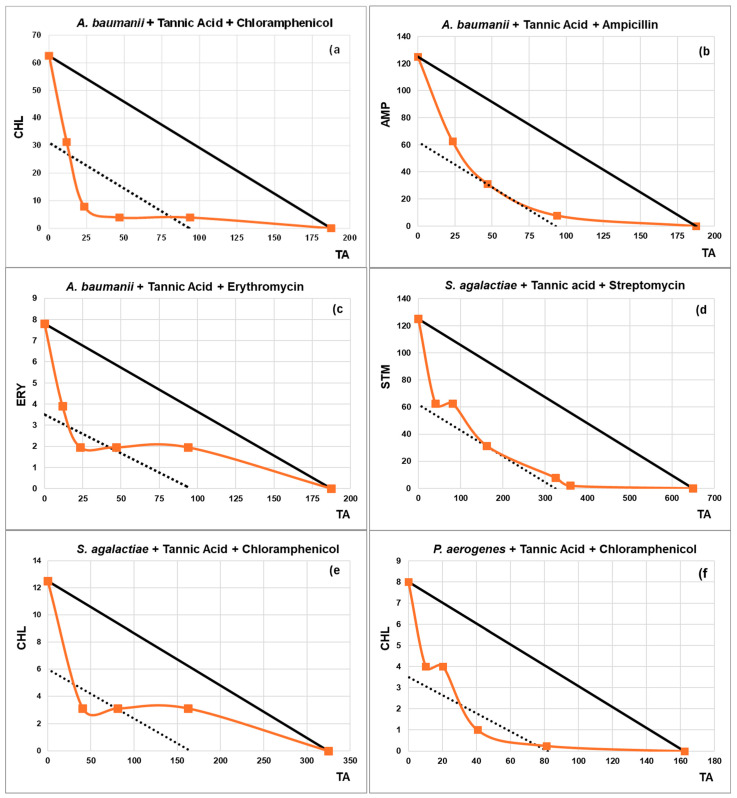
Isobolograms illustrate the interactions of Tannic Acid (TA) with the antibiotics (ABXs) where synergy was detected. The title of each subgraph indicates the bacterium studied along with the composition of the synergy showed in the isobologram.The x-axis represents TA concentrations, while the y-axis represents antibiotic concentrations. The solid line, known as the ‘addition line’, helps differentiate between additive effects—where points fall on or near this line—and synergistic effects, where concave isoboles are found below it. Additionally, there is a dashed line indicating the boundary of synergy. Points situated above or below this dashed line signify different degrees of synergistic interaction. Concentrations of ABXs and TA in (µg/mL).

**Figure 2 plants-13-02717-f002:**
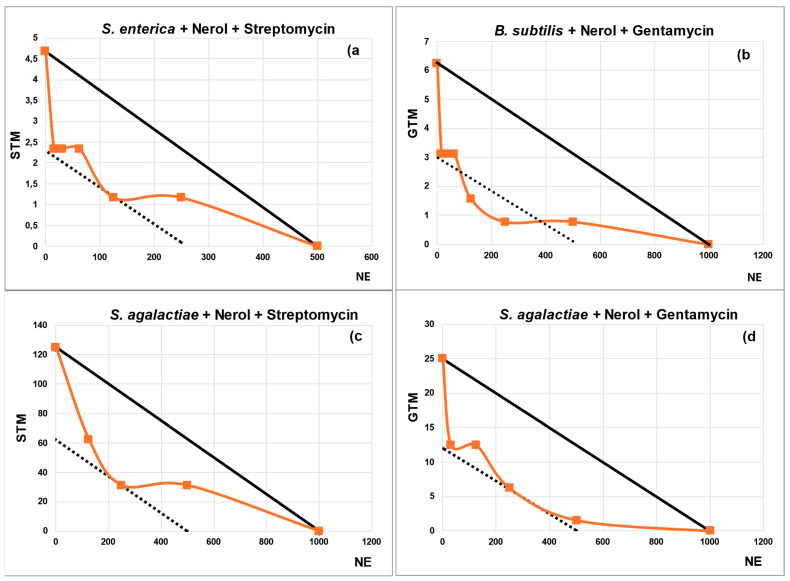
Isobolograms illustrate the interactions of Nerol (NE) with the antibiotics (ABXs) where synergy was detected. The title of each subgraph indicates the bacterium studied along with the composition of the synergy described in the isobologram.The x-axis represents NE concentrations, while the y-axis represents ABX concentrations. The solid line, known as the ‘addition line,’ helps differentiate between additive effects—where points fall on or near this line—and synergistic effects, where concave isoboles are found below it. Additionally, there is a dashed line indicating the boundary of synergy. Points situated above or below this dashed line signify different degrees of synergistic interaction. Concentrations of ABXs and NE in (µg/mL).

**Figure 3 plants-13-02717-f003:**
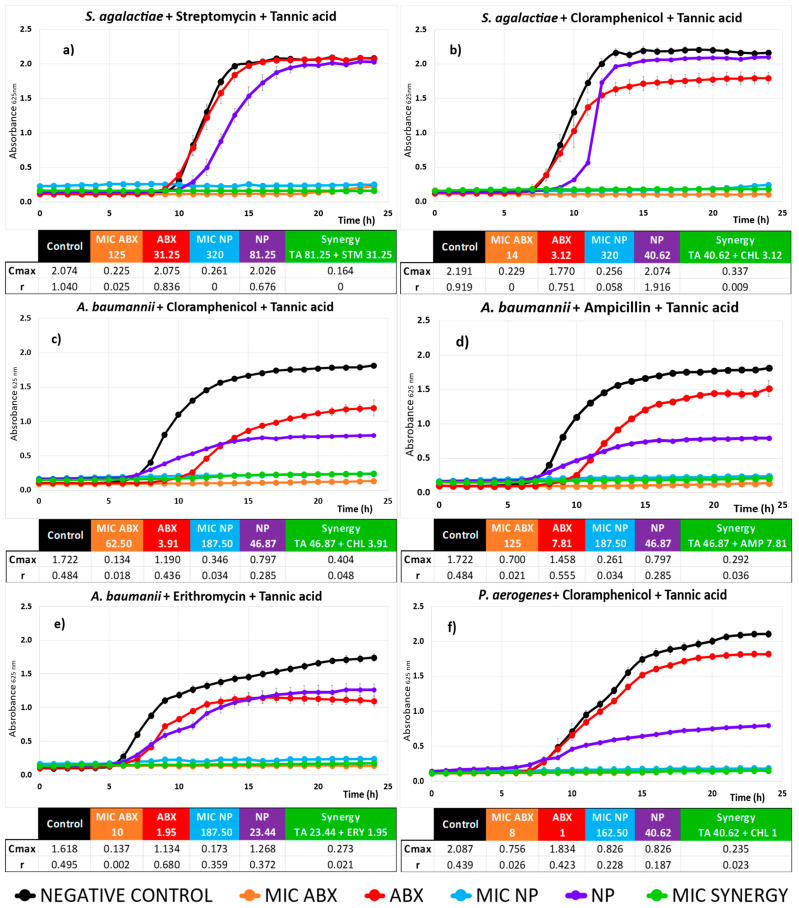
Kinetic assay and Cmax and r values of Tannic Acid (TA) as natural products (NPs), antibiotics (ABXs), and their combinations against different bacteria. The title of each subgraph indicates the bacteria studied along with the composition of the synergy showed in the graph. Black represents the negative control, red represents the ABX concentration in the synergy, purple represents the NP concentration in the synergy, orange represents the Minimum Inhibitory Concentration (MIC) concentration of the ABX, and light blue represents the MIC concentration of the NP. The synergy is represented in green. Error bars indicate standard deviations (*n* = 4). Concentrations of ABXs and TA in (µg/mL). The color code used is described at the end of the chart.

**Figure 4 plants-13-02717-f004:**
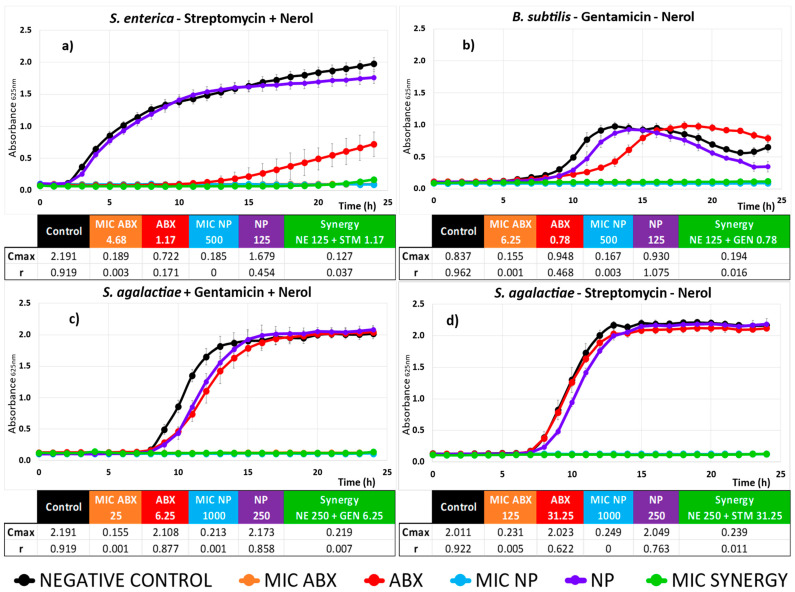
Kinetic assay and Cmax and r values of Nerol (NE) as natural products (NPs), antibiotics (ABXs), and their combinations against different bacteria. The title of each subgraph indicates the bacterium studied along with the composition of the synergy showed in the graph. Black represents the negative control, red represents the ABX concentration in the synergy, purple represents the NP concentration in the synergy, orange represents the Minimum Inhibitory Concentration MIC concentration of the ABX, and light blue represents the MIC concentration of the NP. The synergy is represented in green. Error bars indicate standard deviations (n = 4). Concentrations of ABXs and NE in (µg/mL). The color code used is described at the end of the chart.

**Figure 5 plants-13-02717-f005:**
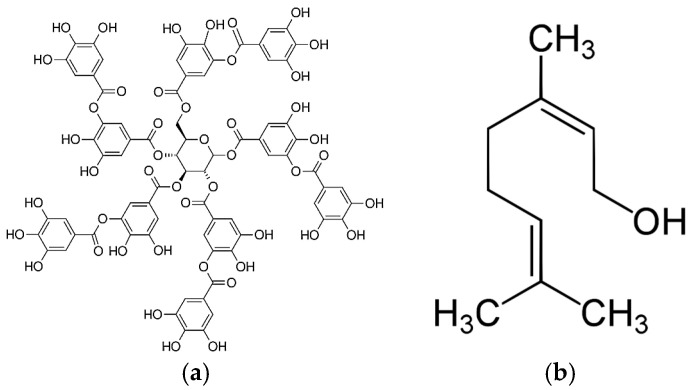
Chemical structure of Tannic Acid (**a**) and Nerol (**b**).

**Table 1 plants-13-02717-t001:** Antimicrobial activity of Nerol and Tannic Acid on selected pathogenic bacteria (µg/mL).

Microorganism	Nerol			Tannic Acid		
	MIC	MBC	MBC/MIC	MIC	MBC	MBC/MIC
*Escherichia coli* (ATCC 25922)	2000	2000	1	>2600	>2600	-
*Salmonella enterica* (ATCC 13311)	500	500	1	>2600	>2600	-
*Klebsiella pneumoniae* (C6)	1000	1000	1	>1387.50	>1387.50	-
*Serratia marcescens* (ATCC 13880)	>1000	>1000	-	>2600	>2600	-
*Proteus mirabilis* (ATCC 35659)	-	-	-	-	-	-
*Pseudomona aeruginosa* (ATCC 27853)	>1000	>1000	-	>2600	>2600	-
*Klebsiella aerogenes* (ATCC 13048)	2000	2000	1	>5200	>5200	-
*Acinetobacter baumannii* (ATCC 19606)	1000	1000	1	187.50	>187.50	>1
*Bacillus subtilis* (ATCC 6633)	500	>1000	>1	>3570	>3570	-
*Staphylococcus aureus* (ATCC 9144)	1000	1000	1	325	>325	>1
*Enterococcus faecalis* (ATCC 19433)	1000	1000	1	650	650	1
*Streptococcus agalactiae* (ATCC 12386)	1000	500	2	320	>320	>1
*Pasteurella aerogenes* (ATCC 27883)	500	>500	>1	162.50	>325	-
*Candida albicans* (ATCC 10231)	1000	1000	1	1800	>1800	>1

MIC, Minimum Inhibitory Concentration; MBC, Minimum Bactericidal Concentration; ATCC, (American Type Culture Collection).

**Table 2 plants-13-02717-t002:** FIC values of Nerol and ABXs combinations.

	ABX	NE in Combination	ABX in Combination	FIC	Interpretation	ABX Reduction (%)	NP Reduction (%)
*Escherichia coli* (ATCC 25922)	STM	250	4.69	0.62	Additivity	50.00	75.00
	GTM	31.25	12.50	1.02	Additivity	0	98.44
	ERY	1000	150	1	Additivity	50.00	50.00
*Salmonella enterica* (ATCC 13311)	AMP	250	7.81	1	Additivity	50.00	50.00
	AMO	15.62	31.25	0.53	Additivity	50.00	96.87
	**STM**	**125**	**1.17**	**0.50**	**Synergy**	**75.00**	**75.00**
	ERY	500	1.17	1.03	Additivity	96.87	0
*Klebsiella pneumoniae* (C6)	AMP	15.62	62.50	1.02	Additivity	0	98.44
	AMO	15.62	250	1.02	Additivity	0	98.44
	STM	125	4.69	0.62	Additivity	50	87.50
	ERY	15.62	18.75	1.02	Additivity	0	98.44
*Klebsiella aerogenes* (ATCC 13048)	GTM	31.25	6.25	1.02	Additivity	0	98.44
	ERY	31.25	75	1.02	Additivity	0	98.44
	CHL	31.25	3.91	1.02	Additivity	0	98.44
	STM	1000	0.78	2.50	Additivity	50	50
*Acinetobacter baumannii* (ATCC 19606)	AMP	500	62.50	0.75	Additivity	50	75
	AMO	15.62	125	1.02	Additivity	0	98.44
	STM	62.50	25	0.62	Additivity	50	87.50
	ERY	125	3.12	0.75	Additivity	50	75
	CHL	250	31.25	0.75	Additivity	75	50
*Bacillus subtilis* (ATCC 6633)	STM	62.50	3.12	0.56	Additivity	50	93.75
	**GTM**	**125**	**0.78**	**0.37**	**Synergy**	**87.50**	**87.50**
	CHL	500	3.91	1.50	Additivity	0	50
*Staphylococcus aureus* (ATCC 9144)	STM	500	75	1.50	Additivity	0	50
	GTM	500	50	1.50	Additivity	0	50
	CHL	500	7.50	1.50	Additivity	0	50
*Enterococcus faecalis* (ATCC 19433)	STM	500	1.56	0.75	Additivity	75	50
	AMP	500	15.62	0.75	Additivity	75	50
	ERY	500	6.25	1.50	Additivity	0	50
*Streptococcus agalactiae* (ATCC 12386)	AMO	500	7.81	1.02	Additivity	50	50
	**STM**	**250**	**31.25**	**0.50**	**Synergy**	**75**	**75**
	**GTM**	**250**	**6.25**	**0.50**	**Synergy**	**75**	**75**
*Pasteurella aerogenes* (ATCC 27883)	STM	500	3.12	0.62	Additivity	87.50	50
	TC	250	1.56	0.75	Additivity	50	75
	GTM	250	3.91	0.75	Additivity	50	75

Concentrations of antibiotic (ABX) and Nerol (NE) in (µg/mL). STM (Streptomycin), GTM (Gentamicin), TC (Tetracycline), AMO (Amoxicillin), ERY (Erythromycin), AMP (Ampicillin), CHL (Chloramphenicol), FIC (fractional inhibitory concentration index), ATCC (American Type Culture Collection), NP (natural product).

**Table 3 plants-13-02717-t003:** FIC values of Tannic Acid and ABXs combinations.

Microorganism	ABX	TA in Combination	ABX in Combination	FIC	Interpretation	ABX Reduction (%)	NP Reduction (%)
*Staphylococcus aureus* (ATCC 9144)	STM	40.62	25	0.75	Additivity	50	87.50
	CHL	40.62	3.75	0.62	Additivity	50	87.50
*Acinetobacter baumannii* (ATCC 19606)	STM	46.87	37.50	0.75	Additivity	50	75
	**CHL**	**46.87**	**3.91**	**0.31**	**Synergy**	**93.75**	**75**
	**AMP**	**46.87**	**7.81**	**0.31**	**Synergy**	**93.75**	**75**
	**ERY**	**23.44**	**1.95**	**0.50**	**Synergy**	**75**	**87.50**
	PEN	23.44	250	1	Additivity	50	75
*Streptococcus agalactiae* (ATCC 12386)	**STM**	**81.25**	**31.25**	**0.50**	**Synergy**	**75**	**75**
	**CHL**	**40.62**	**3.12**	**0.37**	**Synergy**	**75**	**87.50**
	GTM	31.25	25	1.03	Additivity	0	93.75
*Pasteurella aerogenes* (ATCC 27883)	STM	40.62	3.12	0.62	Additivity	87.50	50
	**CHL**	**40.62**	**1**	**0.37**	**Synergy**	**87.50**	**75**
	GTM	40.62	12	1.25	Additivity	0	75

Concentrations of antibiotics (ABX) and Tannic Acid (TA) in (µg/mL). STM (Streptomycin), GTM (Gentamicin), TC (Tetracycline), AMO (Amoxicillin), ERY (Erythromycin), AMP (Ampicillin), CHL (Chloramphenicol), FIC (fractional inhibitory concentration index), ATCC (American Type Culture Collection), NP (natural product).

**Table 4 plants-13-02717-t004:** Chemical details for the antimicrobials used.

Antibiotic/Natural Product	Abbreviation	Chemical Family	CAS-Number	Supplier	Purity	Molecular Weight (g/mol)	Range of Concentrations Tested (µg/mL)
Gentamycin	GTM	Aminoglycosides	1403-66-3	ACO-FARMA	≥97.0%	447.60	250—0.19
Streptomycin	STM		57-92-1		≥97.0%	581.6	400—0.79
Chloramphenicol	CHL	Amphenicols	56-75-7		97.50%	323.1	250—0.23
Amoxicillin	AMO	Β-Lactams	26787-78-0	SIGMA-ALDRICH	96–102%	365.4	500—1.95
Ampicillin	AMP		69-53-4		≥90.00%	394.4	500—7.81
Penicillin G	PEN		69-57-8		96-102%	356.4	1000—7.81
Erythromycin	ERY	Macrolides	114-07-8	ACO-FARMA	95.90%	733.9	600—1.17
Tetracycline	TC	Tetracyclines	64-75-5		99.20%	444.4	100—1.56
Tannic Acid	TA	Polyphenols	1401-55-4		99.00%	1701.2	2600—10.16
Nerol	NE	Monoterpenes	106-25-2		99.00%	154.2	2000—15.62

## Data Availability

Data are contained within the article and Appendix A.

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
