# Peer review of "Enhancing Antibiotic Efficacy with Natural Compounds: Synergistic Activity of Tannic Acid and Nerol with Commercial Antibiotics against Pathogenic Bacteria"

_plants, 2024, doi:10.3390/plants13192717_

Round 1

Reviewer 1 Report

Comments and Suggestions for Authors

The manuscript by Guillermo et al, talks about the synergistic study of natural products and existing antibiotics. The results are quite interesting and the authors have done a good job in the research design.  For the purpose of clarity, for the additive results can the author provide scientific explanation in the discussion section the possible reason for these results?

In addition, most of the references used this study are very old. The authors should improve the references and not use references that are more than 10 years old except when is absolutely necessary. 

Comments on the Quality of English Language

 Minor editing of English language required

Author Response

Question 1. For the purpose of clarity, for the additive results can the author provide scientific explanation in the discussion section the possible reason for these results?

Answer 1. The difference between a synergy and an admixture implies that in synergy the combination between antibiotic and natural product is more effective than their individual parts.

As discussed in detail, one possible mechanism explaining the observed synergies is that the natural product may alter bacterial membranes, facilitating the antibiotic's access to its cytoplasmic target and thereby amplifying the antibiotic's action. However, there is limited information regarding the mechanisms of action behind the additive effects in relation to synergies (Yang et al., 2017).

One possible explanation could be that, in our case, for the additive effects observed with TA and NE, the natural product slightly damages the membrane or causes some intracellular damage, which by itself does not facilitate the antibiotic's action but simply adds to the damage inflicted on the bacteria. Additionally, due to the broad, nonspecific mechanisms associated with natural products, there may not be an opportunity for the combined activities of these compounds to exceed the sum of their partes, as has been suggested in the case of disinfectant combinations (Noel et al., 2021).

However, another study (Yang et al., 2017) argues that the effects on the membrane from the additive interaction of cinnamon bark oil and meropenem are very similar to those observed in previously reported synergistic combinations, indicating that further studies are indeed necessary to clarify this point.

In any case, you are correct that we did not address this point in sufficient detail in the article, so an explanation has been added in the discussion section (Lines 398-417).

 “Finally, it is worth noting that although our study has focused on synergies, given that additive interactions may not be as effective as synergistic interactions, the latter are far more numerous in our results and therefore deserve attention.

As seen in Tables 2 and 3, they also allow for a reduction in the concentration of ABX, and in many cases, the concentration of the adjuvant needed to achieve additivity in combinatory treatment might even be lower than what is observed in synergistic interactions.

Although the mechanisms of additive activities have been little studied, we hypothesize that one possible cause could be that the action of the natural product only slightly damages the membrane or causes some intracellular damage, which by itself does not facilitate the antibiotic's action but simply adds to the damage inflicted on the bacteria. Additionally, due to the broad, nonspecific mechanisms associated with natural products, there may not be an opportunity for the combined activities of these compounds to exceed the sum of their parts, as has been suggested in the case of disinfectant combinations (Noel et al., 2021). However, another study (Yang et al., 2017) argues that the effects on the membrane from the additive interaction of cinnamon bark oil and meropenem are very similar to those observed in previously reported synergistic combinations, indicating that further studies are indeed necessary to clarify this point. It is important to assess the therapeutic potential of additive interactions alongside synergistic ones, as many studies on natural antibiotic adjuvants, including this one, report an equal or greater number of additive interactions (Chovanova et al., 2013; Lederer et al., 2019; Si et al., 2008; van Vuuren et al., 2009; Yang et al., 2017; Yap et al., 2013).

Question 2. In addition, most of the references used this study are very old. The authors should improve the references and not use references that are more than 10 years old except when is absolutely necessary.

Answer 2. The authors have made the following changes to the bibliography as recommended by the reviewers in the introduction:

  • Line 38. The reference (Abreu et al., 2012) has been replaced by (Keck et al., 2024).
  • Line 41. The reference (Betoni et al., 2006) has been replaced by (Lupia et al., 2024).
  • Line 42. The references (Coutinho et al., 2010) and (Sibanda and Okoh, 2007) have been replaced by (Yeshi et al., 2022) and (Jha and Sit, 2022).
  • Line 43. The reference (Basile et al., 1999) has been replaced by (Ahmad et al., 2024) and (Othman et al., 2019).
  • Line 44. The references (Chan et al., 2013) and (Wagner and Ulrich-Merzenich, 2009) have been replaced by (Ferrando et al., 2024) and (Vaou et al., 2022).
  • Line 45. The reference (Taylor, 2013) has been replaced by (Assane et al., 2021) and (Drioiche et al., 2024).
  • Line 45-46. The reference (Abreu et al., 2012) has been replaced by (Guo et al., 2021) and (Zych et al., 2024).
  • Line 49. The references (Taylor, 2013) and (Turek and Stintzing, 2013) have been replaced by (Kumar et al., 2020) and (McEwen and Collignon, 2018).
  • Line 51. The reference (Kalemba and Kunicka, 2003) has been replaced by (Masyita et al., 2022) and (Vaou et al., 2021).
  • Line 52. The reference (Iacobellis et al., 2005) has been replaced by (Daud et al., 2022) and (de Oliveira et al., 2020).
  • Line 61. The reference (Lapczynski et al., 2008) has been replaced by (Api et al., 2023).
  • Line 65. The references (Kotan et al., 2007), (Leonard et al., 2010) have been replaced by (Kasthuri et al., 2022) and (Wang et al., 2020).
  • Line 69. The references (Chung et al., 1998), (Gulcin et al., 2010), (Lopes et al., 1999), (Wu et al., 2004) and (King and Young, 1999) have been replaced by (Guo et al., 2021) and (Jing et al., 2022).
  • Line 87. The citation (Olson et al., 2002) has been replaced by (Macia et al., 2014)

The authors have reorganised the discussion, following the recommendations of the reviewers. In the discussion all bibliographic citations are from the last ten years except in those where it is absolutely necessary.  Additionally, in response to the referees, new updates have also been added during this period.

  • Line 482. The following citations have been included:(Du et al., 2023) and(Li et al., 2024).
  • Line 483-484. The following citations have been included:(Orlowski et al., 2018) and(Park et al., 2017).
  • Line 483. The following citation has been included.(Orlowski et al., 2018).
  • Line 486. The following citation has been included.(Park et al., 2017)

Question 3. Comments on the Quality of English Language. Minor editing of English language required.

Answer 3. The English in both, the existing text and the newly modified and added paragraphs, has been reviewed. Some of the changes are indicated below as an example:

  • Line 46. Corrected environment to environmental.
  • Line 99. Corrected solubilise to solubilize.
  • Line 102. The verb tense has been changed to past tense "the lowest value was".
  • Line 102. The preposition "on" has been changed to "for".
  • Line 103. The verb tense has been changed to past tense. "Ne had MICs".
  • Line 124. The sentence "which had a reduction..." has been modified.
  • Line 131. The sentence has been improved "while STM".
  • Lines 252-255. Verb tenses have been changed to gerund "inhibiting, disrupting, chelting, inactivating".
  • Line 267. The sentence has been improved by rewriting it as "development of pores".
  • Line 459. The spelling has been corrected by introducing the verb "also be applied".
  • Line 543. Substituted adequate for appropriate.
  • Lines 558-559. The verb tenses have been corrected to "were" instead of "was".

New references:

Abreu AC, McBain AJ, Simoes M. Plants as sources of new antimicrobials and resistance-modifying agents. Natural Product Reports 2012; 29: 1007-1021.

Administration FaD. Food additives permitted for direct addiction to food for human consumption. In: Home. F, editor. 21, 2024.

Ahmad MF, Alsayegh AA, Ahmad FA, Akhtar MS, Alavudeen SS, Bantun F, et al. <i>Ganoderma</i><i> lucidum:</i> Insight into antimicrobial and antioxidant properties with development of secondary metabolites. Heliyon 2024; 10.

Api A, Belsito D, Botelho D, Bruze M, Burton G, Cancellieri M, et al. RIFM fragrance ingredient safety assessment, nerol, CAS Registry Number 106-25-2. FOOD AND CHEMICAL TOXICOLOGY 2023; 179.

Assane IM, Santos NA, de Sousa EL, Brasil M, Cilli EM, Pilarski F. Cytotoxicity and antimicrobial activity of synthetic peptides alone or in combination with conventional antimicrobials against fish pathogenic bacteria. Journal of Applied Microbiology 2021; 131: 1762-1774.

Basile A, Giordano S, López-Sáez JA, Cobianchi RC. Antibacterial activity of pure flavonoids isolated from mosses. Phytochemistry 1999; 52: 1479-1482.

Betoni JEC, Passarelli Mantovani R, Nunes Barbosa L, Di Stasi LC, Fernandes Junior A. Synergism between plant extract and antimicrobial drugs used on Staphylococcus aureus diseases. Memorias Do Instituto Oswaldo Cruz 2006; 101: 387-390.

Caprarulo V, Giromini C, Rossi L. Review: Chestnut and quebracho tannins in pig nutrition: the effects on performance and intestinal health. Animal 2021; 15.

Chan BCL, Ip M, Gong H, Lui SL, See RH, Jolivalt C, et al. Synergistic effects of diosmetin with erythromycin against ABC transporter over-expressed methicillin-resistant <i>Staphylococcus aureus</i> (MRSA) RN4220/pUL5054 and inhibition of MRSA pyruvate kinase. Phytomedicine 2013; 20: 611-614.

Chovanova R, Mikulasova M, Vaverkova S. In Vitro Antibacterial and Antibiotic Resistance Modifying Effect of Bioactive Plant Extracts on Methicillin-Resistant Staphylococcus epidermidis. International journal of microbiology 2013; 2013: 760969-760969.

Chung KT, Wong TY, Wei CI, Huang YW, Lin Y. Tannins and human health: A review. Critical Reviews in Food Science and Nutrition 1998; 38: 421-464.

Conidi C, Donato L, Algieri C, Cassano A. Valorization of chestnut processing by-products: A membrane-assisted green strategy for purifying valuable compounds from shells. JOURNAL OF CLEANER PRODUCTION 2022; 378.

Coutinho HDM, Costa JGM, Falcao-Silva VS, Siqueira JP, Jr., Lima EO. Effect of Momordica charantia L. in the resistance to aminoglycosides in methicilin-resistant Staphylococcus aureus. Comparative Immunology Microbiology and Infectious Diseases 2010; 33: 467-471.

Cushnie TPT, Cushnie B, Lamb AJ. Alkaloids: An overview of their antibacterial, antibiotic-enhancing and antivirulence activities. International Journal of Antimicrobial Agents 2014; 44: 377-386.

Daud NM, Putra NR, Jamaludin R, Norodin NSM, Sarkawi NS, Hamzah MHS, et al. Valorisation of plant seed as natural bioactive compounds by various extraction methods: A review. Trends in Food Science & Technology 2022; 119: 201-214.

de Oliveira MS, da Cruz JN, da Costa WA, Silva SG, Brito MdP, Fernandes de Menezes SA, et al. Chemical Composition, Antimicrobial Properties of<i>Siparuna guianensis</i>Essential Oil and a Molecular Docking and Dynamics Molecular Study of its Major Chemical Constituent. Molecules 2020; 25.

Drioiche A, Baammi S, Zibouh K, Al Kamaly O, Alnakhli AM, Remok F, et al. A Study of the Synergistic Effects of Essential Oils from <i>Origanum compactum</i> and <i>Origanum elongatum</i> with Commercial Antibiotics against Highly Prioritized Multidrug-Resistant Bacteria for the World Health Organization. Metabolites 2024; 14.

Du Y, Huo Y, Yang Q, Han Z, Hou L, Cui B, et al. Ultrasmall iron-gallic acid coordination polymer nanodots with antioxidative neuroprotection for PET/MR imaging-guided ischemia stroke therapy. Exploration 2023; 3.

Ferrando N, Pino-Otin MR, Ballestero D, Lorca G, Terrado EM, Langa E. Enhancing Commercial Antibiotics with Trans-Cinnamaldehyde in Gram-Positive and Gram-Negative Bacteria: An In Vitro Approach. Plants-Basel 2024; 13.

Friedman M, Henika PR, Mandrell RE. Bactericidal activities of plant essential oils and some of their isolated constituents against <i>Campylobacter jejuni</i>, <i>Escherichia coli</i>, <i>Listeria monocytogenes</i>, and <i>Salmonella enterica</i>. Journal of Food Protection 2002; 65: 1545-1560.

Gan C, Langa E, Valenzuela A, Ballestero D, Rosa Pino-Otin M. Synergistic Activity of Thymol with Commercial Antibiotics against Critical and High WHO Priority Pathogenic Bacteria. Plants-Basel 2023; 12.

Gulcin I, Huyut Z, Elmastas M, Aboul-Enein HY. Radical scavenging and antioxidant activity of tannic acid. Arabian Journal of Chemistry 2010; 3: 43-53.

Guo Z, Xie W, Lu J, Guo X, Xu J, Xu W, et al. Tannic acid-based metal phenolic networks for bio-applications: a review. JOURNAL OF MATERIALS CHEMISTRY B 2021; 9: 4098-4110.

Iacobellis NS, Lo Cantore P, Capasso F, Senatore F. Antibacterial activity of <i>Cuminum cyminum</i> L. and <i>Carum carvi</i> L. essential oils. Journal of Agricultural and Food Chemistry 2005; 53: 57-61.

Jha AK, Sit N. Extraction of bioactive compounds from plant materials using combination of various novel methods: A review. Trends in Food Science & Technology 2022; 119: 579-591.

Jing W, Xiaolan C, Yu C, Feng Q, Haifeng Y. Pharmacological effects and mechanisms of tannic acid. BIOMEDICINE & PHARMACOTHERAPY 2022; 154.

Kalemba D, Kunicka A. Antibacterial and antifungal properties of essential oils. Current Medicinal Chemistry 2003; 10: 813-829.

Kasthuri T, Swetha TK, Bhaskar JP, Pandian SK. Rapid-killing efficacy substantiates the antiseptic property of the synergistic combination of carvacrol and nerol against nosocomial pathogens. Archives of Microbiology 2022; 204.

Keck JM, Viteri A, Schultz J, Fong R, Whitman C, Poush M, et al. New Agents Are Coming, and So Is the Resistance. Antibiotics-Basel 2024; 13.

King A, Young G. Characteristics and occurrence of phenolic phytochemicals. Journal of the American Dietetic Association 1999; 99: 213-218.

Kotan R, Kordali S, Cakir A. Screening of antibacterial activities of twenty-one oxygenated monoterpenes. Zeitschrift Fur Naturforschung Section C-a Journal of Biosciences 2007; 62: 507-513.

Kumar S, Mukherjee A, Dutta J. Chitosan based nanocomposite films and coatings: Emerging antimicrobial food packaging alternatives. Trends in Food Science & Technology 2020; 97: 196-209.

Langeveld WT, Veldhuizen EJA, Burt SA. Synergy between essential oil components and antibiotics: a review. Critical Reviews in Microbiology 2014; 40: 76-94.

Lapczynski A, Foxenberg RJ, Bhatia SP, Letizia CS, Api AM. Fragrance material review on nerol. Food and Chemical Toxicology 2008; 46: S241-S244.

Lederer S, Dijkstra TMH, Heskes T. Additive Dose Response Models: Defining Synergy. Frontiers in Pharmacology 2019; 10.

Leonard CM, Virijevic S, Regnier T, Combrinck S. Bioactivity of selected essential oils and some components on <i>Listeria monocytogenes</i> biofilms. South African Journal of Botany 2010; 76: 676-680.

Li Y, Duan Y, Li Y, Gu Y, Zhou L, Xiao Z, et al. Cascade loop of ferroptosis induction and immunotherapy based on metal-phenolic networks for combined therapy of colorectal cancer. Exploration 2024.

Lopes GKB, Schulman HM, Hermes-Lima M. Polyphenol tannic acid inhibits hydroxyl radical formation from Fenton reaction by complexing ferrous ions. Biochimica Et Biophysica Acta-General Subjects 1999; 1472: 142-152.

Lupia C, Castagna F, Bava R, Naturale MD, Zicarelli L, Marrelli M, et al. Use of Essential Oils to Counteract the Phenomena of Antimicrobial Resistance in Livestock Species. Antibiotics-Basel 2024; 13.

Macia MD, Rojo-Molinero E, Oliver A. Antimicrobial susceptibility testing in biofilm-growing bacteria. Clinical Microbiology and Infection 2014; 20: 981-990.

Masyita A, Sari RM, Astuti AD, Yasir B, Rumata NR, Bin Emran T, et al. Terpenes and terpenoids as main bioactive compounds of essential oils, their roles in human health and potential application as natural food preservatives. Food Chemistry-X 2022; 13.

McEwen SA, Collignon PJ. Antimicrobial Resistance: a One Health Perspective. In: Schwarz S, Cavaco LM, Shen J, editors. Antimicrobial Resistance in Bacteria from Livestock and Companion Animals, 2018, pp. 521-547.

Myszka K, Schmidt MT, Majcher M, Juzwa W, Olkowicz M, Czaczyk K. Inhibition of <i>quorum sensing</i>-related biofilm of <i>Pseudomonas fluorescens</i> KM121 by <i>Thymus vulgare</i> essential oil and its major bioactive compounds. International Biodeterioration & Biodegradation 2016; 114: 252-259.

Noel DJ, Keevil CW, Wilks SA. Synergism versus Additivity: Defining the Interactions between Common Disinfectants. Mbio 2021; 12.

Olson ME, Ceri H, Morck DW, Buret AG, Read RR. Biofilm bacteria: formation and comparative susceptibility to antibiotics. Canadian Journal of Veterinary Research-Revue Canadienne De Recherche Veterinaire 2002; 66: 86-92.

Orlowski P, Zmigrodzka M, Tomaszewska E, Ranoszek-Soliwoda K, Czupryn M, Antos-Bielska M, et al. Tannic acid-modified silver nanoparticles for wound healing: the importance of size. International Journal of Nanomedicine 2018; 13: 991-1007.

Othman L, Sleiman A, Abdel-Massih RM. Antimicrobial Activity of Polyphenols and Alkaloids in Middle Eastern Plants. Frontiers in Microbiology 2019; 10.

Park JH, Choi S, Moon HC, Seo H, Kim JY, Hong S-P, et al. Antimicrobial spray nanocoating of supramolecular Fe(III)-tannic acid metal-organic coordination complex: applications to shoe insoles and fruits. Scientific Reports 2017; 7.

Pino-Otin MR, Valenzuela A, Gan C, Lorca G, Ferrando N, Langa E, et al. Ecotoxicity of five veterinary antibiotics on indicator organisms and water and soil communities. Ecotoxicology and Environmental Safety 2024; 274.

Ren A, Zhang W, Thomas HG, Barish A, Berry S, Kiel JS, et al. A Tannic Acid-Based Medical Food, Cesinex<SUP>A®</SUP>, Exhibits Broad-Spectrum Antidiarrheal Properties: A Mechanistic and Clinical Study. Digestive Diseases and Sciences 2012; 57: 99-108.

Sharma A, Anurag J, Kaur J, Kesharwani A, Parihar VK. Antimicrobial Potential of Polyphenols: An Update on Alternative for Combating Antimicrobial Resistance. Medicinal Chemistry 2024; 20: 576-596.

Si H, Hu J, Liu Z, Zeng Z-l. Antibacterial effect of oregano essential oil alone and in combination with antibiotics against extended-spectrum β-lactamase-producing <i>Escherichia coli</i>. Fems Immunology and Medical Microbiology 2008; 53: 190-194.

Sibanda T, Okoh AI. The challenges of overcoming antibiotic resistance: Plant extracts as potential sources of antimicrobial and resistance modifying agents. African Journal of Biotechnology 2007; 6: 2886-2896.

Stumpf S, Hostnik G, Primozic M, Leitgeb M, Salminen J-P, Bren U. The Effect of Growth Medium Strength on Minimum Inhibitory Concentrations of Tannins and Tannin Extracts against<i>E. coli</i>. Molecules 2020; 25.

Taylor PW. Alternative natural sources for a new generation of antibacterial agents. International Journal of Antimicrobial Agents 2013; 42: 195-201.

Teuber M. Veterinary use and antibiotic resistance. Current Opinion in Microbiology 2001; 4: 493-499.

Turek C, Stintzing FC. Stability of Essential Oils: A Review. Comprehensive Reviews in Food Science and Food Safety 2013; 12: 40-53.

Union E. Regulation (EU) 2019/6 of the European Parliament and of the Council of 11 December 2018 on veterinary medicinal products and repealing Directive 2001/82/EC, 2019.

van Vuuren SF, Suliman S, Viljoen AM. The antimicrobial activity of four commercial essential oils in combination with conventional antimicrobials. Letters in Applied Microbiology 2009; 48: 440-446.

Vaou N, Stavropoulou E, Voidarou C, Tsakris Z, Rozos G, Tsigalou C, et al. Interactions between Medical Plant-Derived Bioactive Compounds: Focus on Antimicrobial Combination Effects. Antibiotics-Basel 2022; 11.

Vaou N, Stavropoulou E, Voidarou C, Tsigalou C, Bezirtzoglou E. Towards Advances in Medicinal Plant Antimicrobial Activity: A Review Study on Challenges and Future Perspectives. Microorganisms 2021; 9.

Verma A, Srivastava R, Sonar P, Yadav R. Traditional, phytochemical, and biological aspects of <i>Rosa alba</i> L.: a systematic review. FUTURE JOURNAL OF PHARMACEUTICAL SCIENCES 2020; 6.

Wagner H, Ulrich-Merzenich G. Synergy research: Approaching a new generation of phytopharmaceuticals. Phytomedicine 2009; 16: 97-110.

Wang Z, Yang K, Chen L, Yan R, Qu S, Li Y-x, et al. Activities of Nerol, a natural plant active ingredient, against <i>Candida albicans</i> in vitro and in vivo. Applied Microbiology and Biotechnology 2020; 104: 5039-5052.

Wu LT, Chu CC, Chung JG, Chen CH, Hsu LS, Liu JK, et al. Effects of tannic acid and its related compounds on food mutagens or hydrogen peroxide-induced DNA strands breaks in human lymphocytes. Mutation Research-Fundamental and Molecular Mechanisms of Mutagenesis 2004; 556: 75-82.

Yang S-K, Yusoff K, Mai C-W, Lim W-M, Yap W-S, Lim S-HE, et al. Additivity vs. Synergism: Investigation of the Additive Interaction of Cinnamon Bark Oil and Meropenem in Combinatory Therapy. Molecules 2017; 22.

Yap PSX, Lim SHE, Hu CP, Yiap BC. Combination of essential oils and antibiotics reduce antibiotic resistance in plasmid-conferred multidrug resistant bacteria. Phytomedicine 2013; 20: 710-713.

Yeshi K, Crayn D, Ritmejeryte E, Wangchuk P. Plant Secondary Metabolites Produced in Response to Abiotic Stresses Has Potential Application in Pharmaceutical Product Development. Molecules 2022; 27.

Zych S, Adaszynska-Skwirzynska M, Szewczuk MA, Szczerbinska D. Interaction between Enrofloxacin and Three Essential Oils (Cinnamon Bark, Clove Bud and Lavender Flower)-A Study on Multidrug-Resistant <i>Escherichia coli</i> Strains Isolated from 1-Day-Old Broiler Chickens. International Journal of Molecular Sciences 2024; 25.

Reviewer 2 Report

Comments and Suggestions for Authors

The studies entitled “Enhancing Antibiotic Efficacy with Natural Compounds: Synergistic Activity of Tannic Acid and Nerol with Commercial Antibiotics Against Pathogenic Bacteria” investigated the antimicrobial properties of natural compounds, particularly tannic acid and nerol, in combination with commercial antibiotics to combat bacterial infections and antimicrobial resistance. The research highlights the effectiveness of these natural products against various bacterial strains, demonstrating their potential to enhance antibiotic efficacy while reducing required doses. Key findings include significant reductions in minimum inhibitory concentrations (MICs) when combined with antibiotics, as well as insights into their mechanisms of action, such as disruption of bacterial membranes and interference with protein synthesis. The article is well organized, which is suggested to accept after addressing the following minor issues:

1. Author should reorganize the figure label.

2. TA is a type of polyphenol which can form metal complex through coordination. It is interested to see the author discuss or study on the antibacterial effects of polyphenolic metallic-complexes like following articles: https://doi.org/10.1002/EXP.20220041

https://doi.org/10.1002/EXP.20230117

3. It is suggested to reduce tables and add more figures to support the data from the table. 

Comments on the Quality of English Language

Minor editing of English language required to eliminate grammar mistakes and typos.

Author Response

Open Review

(x) I would not like to sign my review report

( ) I would like to sign my review report

Quality of English Language

( ) I am not qualified to assess the quality of English in this paper.

( ) The English is very difficult to understand/incomprehensible.

( ) Extensive editing of English language required.

( ) Moderate editing of English language required.

(x) Minor editing of English language required.

( ) English language fine. No issues detected.

The studies entitled “Enhancing Antibiotic Efficacy with Natural Compounds: Synergistic Activity of Tannic Acid and Nerol with Commercial Antibiotics Against Pathogenic Bacteria” investigated the antimicrobial properties of natural compounds, particularly tannic acid and nerol, in combination with commercial antibiotics to combat bacterial infections and antimicrobial resistance. The research highlights the effectiveness of these natural products against various bacterial strains, demonstrating their potential to enhance antibiotic efficacy while reducing required doses. Key findings include significant reductions in minimum inhibitory concentrations (MICs) when combined with antibiotics, as well as insights into their mechanisms of action, such as disruption of bacterial membranes and interference with protein synthesis. The article is well organized, which is suggested to accept after addressing the following minor issues:

Question 1 Author should reorganize the figure label.

Answer 1. Following the reviewer's indications, the figure label in the article have been revised and the following modifications have been made:

  • The titles of all the graphs have been improved by adjusting their size as well as their alignment and placement.
  • Figures 1 and 2: The titles of the isobolograms have been modified to a uniform size.
  • Figures 3 and 4: The colours of the bottom label of the kinetics have been rearranged, and the size of the bottom label has been increased to make it easier to read

Question 2. TA is a type of polyphenol which can form metal complex through coordination. It is interested to see the author discuss or study on the antibacterial effects of polyphenolic metallic-complexes like following articles:

Answer 2. Of course, this point has been commented in the lines 480-487                         of the Discussion section and some references have been added:

“New perspectives in the antimicrobial use of these natural products are emerging. TA, a type of polyphenol, can form metal complexes through coordination, as has been observed with other polyphenols such as gallic acid (Du et al., 2023; Li et al., 2024). TA can form coordination compounds with silver and iron, leading to nanoparticles with antimicrobial activity (Orlowski et al., 2018; Park et al., 2017). This opens the door to potential new antimicrobial applications for these natural products, such as in wound management (Orlowski), or in shoe insoles and fruit preservation (Park et al., 2017). Additionally, the efficacy of these nanoparticles could potentially be enhanced by combining the natural product with an antibiotic to achieve synergistic effects”.

Question 3. It is suggested to reduce tables and add more figures to support the data from the table.

Answer 3. Thank you for the suggestion. We understand that the tables are extensive, especially Tables 3 and 4 (now Tables 2 and 3). However, given the volume of experimental work, we respectfully believe that the current design of these tables effectively presents the set of results in a way that allows for easy consultation and comparison at a glance. This design helps avoid the inconvenience of having to search for results or discussions scattered across different sections. On the other hand, it is challenging to represent the quantitative data from the tables in figures beyond those that have already been designed.

Nonetheless, in response to your suggestion, we have reduced the number of tables so that Table 2: 'Antimicrobial activity of antibiotics (MIC, µg/mL) on pathogenic bacteria tested' has been moved to the Support Information 1.

Additionally, Tables 1, 2, and 3 have been condensed and reformatted to minimize their size and prevent them from occupying too much space in the article.

Question 4. Comments on the Quality of English Language

Answer 4. The English in both, the existing text and the newly modified and added paragraphs, has been reviewed. Some of the changes are indicated below as an example:

  • Line 46. Corrected environment to environmental.
  • Line 99. Corrected solubilise to solubilize.
  • Line 102. The verb tense has been changed to past tense "the lowest value was".
  • Line 102. The preposition "on" has been changed to "for".
  • Line 103. The verb tense has been changed to past tense. "Ne had MICs".
  • Line 124. The sentence "which had a reduction..." has been modified.
  • Line 131. The sentence has been improved "while STM".
  • Lines 252-255. Verb tenses have been changed to gerund "inhibiting, disrupting, chelting, inactivating".
  • Line 267. The sentence has been improved by rewriting it as "development of pores".
  • Line 459. The spelling has been corrected by introducing the verb "also be applied".
  • Line 543. Substituted adequate for appropriate.
  • Lines 558-559. The verb tenses have been corrected to "were" instead of "was".

New references:

Abreu AC, McBain AJ, Simoes M. Plants as sources of new antimicrobials and resistance-modifying agents. Natural Product Reports 2012; 29: 1007-1021.

Administration FaD. Food additives permitted for direct addiction to food for human consumption. In: Home. F, editor. 21, 2024.

Ahmad MF, Alsayegh AA, Ahmad FA, Akhtar MS, Alavudeen SS, Bantun F, et al. <i>Ganoderma</i><i> lucidum:</i> Insight into antimicrobial and antioxidant properties with development of secondary metabolites. Heliyon 2024; 10.

Api A, Belsito D, Botelho D, Bruze M, Burton G, Cancellieri M, et al. RIFM fragrance ingredient safety assessment, nerol, CAS Registry Number 106-25-2. FOOD AND CHEMICAL TOXICOLOGY 2023; 179.

Assane IM, Santos NA, de Sousa EL, Brasil M, Cilli EM, Pilarski F. Cytotoxicity and antimicrobial activity of synthetic peptides alone or in combination with conventional antimicrobials against fish pathogenic bacteria. Journal of Applied Microbiology 2021; 131: 1762-1774.

Basile A, Giordano S, López-Sáez JA, Cobianchi RC. Antibacterial activity of pure flavonoids isolated from mosses. Phytochemistry 1999; 52: 1479-1482.

Betoni JEC, Passarelli Mantovani R, Nunes Barbosa L, Di Stasi LC, Fernandes Junior A. Synergism between plant extract and antimicrobial drugs used on Staphylococcus aureus diseases. Memorias Do Instituto Oswaldo Cruz 2006; 101: 387-390.

Caprarulo V, Giromini C, Rossi L. Review: Chestnut and quebracho tannins in pig nutrition: the effects on performance and intestinal health. Animal 2021; 15.

Chan BCL, Ip M, Gong H, Lui SL, See RH, Jolivalt C, et al. Synergistic effects of diosmetin with erythromycin against ABC transporter over-expressed methicillin-resistant <i>Staphylococcus aureus</i> (MRSA) RN4220/pUL5054 and inhibition of MRSA pyruvate kinase. Phytomedicine 2013; 20: 611-614.

Chovanova R, Mikulasova M, Vaverkova S. In Vitro Antibacterial and Antibiotic Resistance Modifying Effect of Bioactive Plant Extracts on Methicillin-Resistant Staphylococcus epidermidis. International journal of microbiology 2013; 2013: 760969-760969.

Chung KT, Wong TY, Wei CI, Huang YW, Lin Y. Tannins and human health: A review. Critical Reviews in Food Science and Nutrition 1998; 38: 421-464.

Conidi C, Donato L, Algieri C, Cassano A. Valorization of chestnut processing by-products: A membrane-assisted green strategy for purifying valuable compounds from shells. JOURNAL OF CLEANER PRODUCTION 2022; 378.

Coutinho HDM, Costa JGM, Falcao-Silva VS, Siqueira JP, Jr., Lima EO. Effect of Momordica charantia L. in the resistance to aminoglycosides in methicilin-resistant Staphylococcus aureus. Comparative Immunology Microbiology and Infectious Diseases 2010; 33: 467-471.

Cushnie TPT, Cushnie B, Lamb AJ. Alkaloids: An overview of their antibacterial, antibiotic-enhancing and antivirulence activities. International Journal of Antimicrobial Agents 2014; 44: 377-386.

Daud NM, Putra NR, Jamaludin R, Norodin NSM, Sarkawi NS, Hamzah MHS, et al. Valorisation of plant seed as natural bioactive compounds by various extraction methods: A review. Trends in Food Science & Technology 2022; 119: 201-214.

de Oliveira MS, da Cruz JN, da Costa WA, Silva SG, Brito MdP, Fernandes de Menezes SA, et al. Chemical Composition, Antimicrobial Properties of<i>Siparuna guianensis</i>Essential Oil and a Molecular Docking and Dynamics Molecular Study of its Major Chemical Constituent. Molecules 2020; 25.

Drioiche A, Baammi S, Zibouh K, Al Kamaly O, Alnakhli AM, Remok F, et al. A Study of the Synergistic Effects of Essential Oils from <i>Origanum compactum</i> and <i>Origanum elongatum</i> with Commercial Antibiotics against Highly Prioritized Multidrug-Resistant Bacteria for the World Health Organization. Metabolites 2024; 14.

Du Y, Huo Y, Yang Q, Han Z, Hou L, Cui B, et al. Ultrasmall iron-gallic acid coordination polymer nanodots with antioxidative neuroprotection for PET/MR imaging-guided ischemia stroke therapy. Exploration 2023; 3.

Ferrando N, Pino-Otin MR, Ballestero D, Lorca G, Terrado EM, Langa E. Enhancing Commercial Antibiotics with Trans-Cinnamaldehyde in Gram-Positive and Gram-Negative Bacteria: An In Vitro Approach. Plants-Basel 2024; 13.

Friedman M, Henika PR, Mandrell RE. Bactericidal activities of plant essential oils and some of their isolated constituents against <i>Campylobacter jejuni</i>, <i>Escherichia coli</i>, <i>Listeria monocytogenes</i>, and <i>Salmonella enterica</i>. Journal of Food Protection 2002; 65: 1545-1560.

Gan C, Langa E, Valenzuela A, Ballestero D, Rosa Pino-Otin M. Synergistic Activity of Thymol with Commercial Antibiotics against Critical and High WHO Priority Pathogenic Bacteria. Plants-Basel 2023; 12.

Gulcin I, Huyut Z, Elmastas M, Aboul-Enein HY. Radical scavenging and antioxidant activity of tannic acid. Arabian Journal of Chemistry 2010; 3: 43-53.

Guo Z, Xie W, Lu J, Guo X, Xu J, Xu W, et al. Tannic acid-based metal phenolic networks for bio-applications: a review. JOURNAL OF MATERIALS CHEMISTRY B 2021; 9: 4098-4110.

Iacobellis NS, Lo Cantore P, Capasso F, Senatore F. Antibacterial activity of <i>Cuminum cyminum</i> L. and <i>Carum carvi</i> L. essential oils. Journal of Agricultural and Food Chemistry 2005; 53: 57-61.

Jha AK, Sit N. Extraction of bioactive compounds from plant materials using combination of various novel methods: A review. Trends in Food Science & Technology 2022; 119: 579-591.

Jing W, Xiaolan C, Yu C, Feng Q, Haifeng Y. Pharmacological effects and mechanisms of tannic acid. BIOMEDICINE & PHARMACOTHERAPY 2022; 154.

Kalemba D, Kunicka A. Antibacterial and antifungal properties of essential oils. Current Medicinal Chemistry 2003; 10: 813-829.

Kasthuri T, Swetha TK, Bhaskar JP, Pandian SK. Rapid-killing efficacy substantiates the antiseptic property of the synergistic combination of carvacrol and nerol against nosocomial pathogens. Archives of Microbiology 2022; 204.

Keck JM, Viteri A, Schultz J, Fong R, Whitman C, Poush M, et al. New Agents Are Coming, and So Is the Resistance. Antibiotics-Basel 2024; 13.

King A, Young G. Characteristics and occurrence of phenolic phytochemicals. Journal of the American Dietetic Association 1999; 99: 213-218.

Kotan R, Kordali S, Cakir A. Screening of antibacterial activities of twenty-one oxygenated monoterpenes. Zeitschrift Fur Naturforschung Section C-a Journal of Biosciences 2007; 62: 507-513.

Kumar S, Mukherjee A, Dutta J. Chitosan based nanocomposite films and coatings: Emerging antimicrobial food packaging alternatives. Trends in Food Science & Technology 2020; 97: 196-209.

Langeveld WT, Veldhuizen EJA, Burt SA. Synergy between essential oil components and antibiotics: a review. Critical Reviews in Microbiology 2014; 40: 76-94.

Lapczynski A, Foxenberg RJ, Bhatia SP, Letizia CS, Api AM. Fragrance material review on nerol. Food and Chemical Toxicology 2008; 46: S241-S244.

Lederer S, Dijkstra TMH, Heskes T. Additive Dose Response Models: Defining Synergy. Frontiers in Pharmacology 2019; 10.

Leonard CM, Virijevic S, Regnier T, Combrinck S. Bioactivity of selected essential oils and some components on <i>Listeria monocytogenes</i> biofilms. South African Journal of Botany 2010; 76: 676-680.

Li Y, Duan Y, Li Y, Gu Y, Zhou L, Xiao Z, et al. Cascade loop of ferroptosis induction and immunotherapy based on metal-phenolic networks for combined therapy of colorectal cancer. Exploration 2024.

Lopes GKB, Schulman HM, Hermes-Lima M. Polyphenol tannic acid inhibits hydroxyl radical formation from Fenton reaction by complexing ferrous ions. Biochimica Et Biophysica Acta-General Subjects 1999; 1472: 142-152.

Lupia C, Castagna F, Bava R, Naturale MD, Zicarelli L, Marrelli M, et al. Use of Essential Oils to Counteract the Phenomena of Antimicrobial Resistance in Livestock Species. Antibiotics-Basel 2024; 13.

Macia MD, Rojo-Molinero E, Oliver A. Antimicrobial susceptibility testing in biofilm-growing bacteria. Clinical Microbiology and Infection 2014; 20: 981-990.

Masyita A, Sari RM, Astuti AD, Yasir B, Rumata NR, Bin Emran T, et al. Terpenes and terpenoids as main bioactive compounds of essential oils, their roles in human health and potential application as natural food preservatives. Food Chemistry-X 2022; 13.

McEwen SA, Collignon PJ. Antimicrobial Resistance: a One Health Perspective. In: Schwarz S, Cavaco LM, Shen J, editors. Antimicrobial Resistance in Bacteria from Livestock and Companion Animals, 2018, pp. 521-547.

Myszka K, Schmidt MT, Majcher M, Juzwa W, Olkowicz M, Czaczyk K. Inhibition of <i>quorum sensing</i>-related biofilm of <i>Pseudomonas fluorescens</i> KM121 by <i>Thymus vulgare</i> essential oil and its major bioactive compounds. International Biodeterioration & Biodegradation 2016; 114: 252-259.

Noel DJ, Keevil CW, Wilks SA. Synergism versus Additivity: Defining the Interactions between Common Disinfectants. Mbio 2021; 12.

Olson ME, Ceri H, Morck DW, Buret AG, Read RR. Biofilm bacteria: formation and comparative susceptibility to antibiotics. Canadian Journal of Veterinary Research-Revue Canadienne De Recherche Veterinaire 2002; 66: 86-92.

Orlowski P, Zmigrodzka M, Tomaszewska E, Ranoszek-Soliwoda K, Czupryn M, Antos-Bielska M, et al. Tannic acid-modified silver nanoparticles for wound healing: the importance of size. International Journal of Nanomedicine 2018; 13: 991-1007.

Othman L, Sleiman A, Abdel-Massih RM. Antimicrobial Activity of Polyphenols and Alkaloids in Middle Eastern Plants. Frontiers in Microbiology 2019; 10.

Park JH, Choi S, Moon HC, Seo H, Kim JY, Hong S-P, et al. Antimicrobial spray nanocoating of supramolecular Fe(III)-tannic acid metal-organic coordination complex: applications to shoe insoles and fruits. Scientific Reports 2017; 7.

Pino-Otin MR, Valenzuela A, Gan C, Lorca G, Ferrando N, Langa E, et al. Ecotoxicity of five veterinary antibiotics on indicator organisms and water and soil communities. Ecotoxicology and Environmental Safety 2024; 274.

Ren A, Zhang W, Thomas HG, Barish A, Berry S, Kiel JS, et al. A Tannic Acid-Based Medical Food, Cesinex<SUP>A®</SUP>, Exhibits Broad-Spectrum Antidiarrheal Properties: A Mechanistic and Clinical Study. Digestive Diseases and Sciences 2012; 57: 99-108.

Sharma A, Anurag J, Kaur J, Kesharwani A, Parihar VK. Antimicrobial Potential of Polyphenols: An Update on Alternative for Combating Antimicrobial Resistance. Medicinal Chemistry 2024; 20: 576-596.

Si H, Hu J, Liu Z, Zeng Z-l. Antibacterial effect of oregano essential oil alone and in combination with antibiotics against extended-spectrum β-lactamase-producing <i>Escherichia coli</i>. Fems Immunology and Medical Microbiology 2008; 53: 190-194.

Sibanda T, Okoh AI. The challenges of overcoming antibiotic resistance: Plant extracts as potential sources of antimicrobial and resistance modifying agents. African Journal of Biotechnology 2007; 6: 2886-2896.

Stumpf S, Hostnik G, Primozic M, Leitgeb M, Salminen J-P, Bren U. The Effect of Growth Medium Strength on Minimum Inhibitory Concentrations of Tannins and Tannin Extracts against<i>E. coli</i>. Molecules 2020; 25.

Taylor PW. Alternative natural sources for a new generation of antibacterial agents. International Journal of Antimicrobial Agents 2013; 42: 195-201.

Teuber M. Veterinary use and antibiotic resistance. Current Opinion in Microbiology 2001; 4: 493-499.

Turek C, Stintzing FC. Stability of Essential Oils: A Review. Comprehensive Reviews in Food Science and Food Safety 2013; 12: 40-53.

Union E. Regulation (EU) 2019/6 of the European Parliament and of the Council of 11 December 2018 on veterinary medicinal products and repealing Directive 2001/82/EC, 2019.

van Vuuren SF, Suliman S, Viljoen AM. The antimicrobial activity of four commercial essential oils in combination with conventional antimicrobials. Letters in Applied Microbiology 2009; 48: 440-446.

Vaou N, Stavropoulou E, Voidarou C, Tsakris Z, Rozos G, Tsigalou C, et al. Interactions between Medical Plant-Derived Bioactive Compounds: Focus on Antimicrobial Combination Effects. Antibiotics-Basel 2022; 11.

Vaou N, Stavropoulou E, Voidarou C, Tsigalou C, Bezirtzoglou E. Towards Advances in Medicinal Plant Antimicrobial Activity: A Review Study on Challenges and Future Perspectives. Microorganisms 2021; 9.

Verma A, Srivastava R, Sonar P, Yadav R. Traditional, phytochemical, and biological aspects of <i>Rosa alba</i> L.: a systematic review. FUTURE JOURNAL OF PHARMACEUTICAL SCIENCES 2020; 6.

Wagner H, Ulrich-Merzenich G. Synergy research: Approaching a new generation of phytopharmaceuticals. Phytomedicine 2009; 16: 97-110.

Wang Z, Yang K, Chen L, Yan R, Qu S, Li Y-x, et al. Activities of Nerol, a natural plant active ingredient, against <i>Candida albicans</i> in vitro and in vivo. Applied Microbiology and Biotechnology 2020; 104: 5039-5052.

Wu LT, Chu CC, Chung JG, Chen CH, Hsu LS, Liu JK, et al. Effects of tannic acid and its related compounds on food mutagens or hydrogen peroxide-induced DNA strands breaks in human lymphocytes. Mutation Research-Fundamental and Molecular Mechanisms of Mutagenesis 2004; 556: 75-82.

Yang S-K, Yusoff K, Mai C-W, Lim W-M, Yap W-S, Lim S-HE, et al. Additivity vs. Synergism: Investigation of the Additive Interaction of Cinnamon Bark Oil and Meropenem in Combinatory Therapy. Molecules 2017; 22.

Yap PSX, Lim SHE, Hu CP, Yiap BC. Combination of essential oils and antibiotics reduce antibiotic resistance in plasmid-conferred multidrug resistant bacteria. Phytomedicine 2013; 20: 710-713.

Yeshi K, Crayn D, Ritmejeryte E, Wangchuk P. Plant Secondary Metabolites Produced in Response to Abiotic Stresses Has Potential Application in Pharmaceutical Product Development. Molecules 2022; 27.

Zych S, Adaszynska-Skwirzynska M, Szewczuk MA, Szczerbinska D. Interaction between Enrofloxacin and Three Essential Oils (Cinnamon Bark, Clove Bud and Lavender Flower)-A Study on Multidrug-Resistant <i>Escherichia coli</i> Strains Isolated from 1-Day-Old Broiler Chickens. International Journal of Molecular Sciences 2024; 25.

Reviewer 3 Report

Comments and Suggestions for Authors

The manuscript is an in vitro study focusing on the synergy between two natural antimicrobials and different commercial antibiotics against resistant pathogenic bacteria. The topic of the article has some interest for clinical applications, but the manuscript suffers from some negative aspects. The research conducted is a preliminary in vitro study with limited knowledge about the mechanism of action of the biocidal compounds. The discussion section is confusingly written and should be revised. Other specific comments are included below. I recommend major revision of the manuscript before considering it for publication.

Introduction

Lines 30-31. Rewrite the sentences. “As antibiotic consumption has notably increased in the last twenty years, particularly for aminoglycosides [2]. The World Health organization [3] has released a catalogue of drug-resistant bacteria”.

Lines 34-35. Rename microorganisms in italics.

Line 46. Replace “environment” by “environmental”.

Line 60. Rename microorganisms in italics.

Results

Line 105. Replace “de” by “the”.

Lines 104-106. Rewrite the sentence. This statement is only true for some microorganisms and natural products.

Standardise the units “mg/ml” or “mg/mL” throughout the manuscript.

Table 1. Why are MBC values not well defined in cases of good antimicrobial activity?

Tables 2, 3, 4. Include a footnote with the abbreviations mentioned in the table.

Figures. Improve the quality of the figures.

Figure 1. Correct legend, replace NE by TA.

Isobolograms. A more in-depth explanation of the figure and its relevance would be appreciated.

Discussion

It is difficult to follow the discussion section. It is necessary to reorganise and synthesise, as well as to support the explanation with the corresponding tables or figures.

Introduce the discussion of the results. Explain why you choose the natural products.

Justify why you use only one strain of each microorganism.

Explain why natural products do not contribute to antimicrobial resistance.

Is the concentration range of the natural product-antibiotic combination appropriate for clinical use? Include potential applications of the natural compounds in combination with antibiotics.

Materials and methods

As the Materials and methods section appears at the end of the manuscript, all abbreviations should be defined in the Results section.

Clarify the abbreviation for antibiotic, either ATB or ABX.

Line 415. Replace “Table 1” by “Table 5”.

Table 5. Include the range of concentrations tested.

Line 465. Which McFarland standard was used?

Line 478. Replace “tasted” by “tested”.

Section 4.2. Include a table with the details of the tested strains (collection number, media and temperature for the antimicrobial experiments)

Sections 4.3 and 4.4. Reduce these sections by eliminating unnecessary details for the experimental set-up such as microplate configuration.

Author Response

Open Review

(x) I would not like to sign my review report

( ) I would like to sign my review report

Quality of English Language

( ) I am not qualified to assess the quality of English in this paper.

( ) The English is very difficult to understand/incomprehensible.

( ) Extensive editing of English language required.

( ) Moderate editing of English language required.

( ) Minor editing of English language required.

(x) English language fine. No issues detected.

The manuscript is an in vitro study focusing on the synergy between two natural antimicrobials and different commercial antibiotics against resistant pathogenic bacteria. The topic of the article has some interest for clinical applications, but the manuscript suffers from some negative aspects. The research conducted is a preliminary in vitro study with limited knowledge about the mechanism of action of the biocidal compounds. The discussion section is confusingly written and should be revised. Other specific comments are included below. I recommend major revision of the manuscript before considering it for publication.

 Introduction.

QUESTION 1. Lines 30-32. Rewrite the sentences. “As antibiotic consumption has notably increased in the last twenty years, particularly for aminoglycosides [2]. The World Health organization [3] has released a catalogue of drug-resistant bacteria”.

Answer 1. The authors have rewritten the sentences suggested by the reviewer as follows in the lines 30-32:

“In the past two decades, antibiotic consumption has significantly increased, with a particular rise in the use of aminoglycosides. The World Health Organization (WHO) has published a comprehensive list of antibiotic-resistant bacterial pathogens (WHO,2015).”

QUESTION 2. Lines 34-35. Rename microorganisms in italics.

Answer 2. Sorry for the mistake. The names of the microorganisms         indicated have been written in italics in Lines 35-36.

QUESTION 3. Line 46. Replace “environment” by “environmental”.

Answer 3: Following the reviewer's indications, the replacement of                          “environment” by “environmental” in Line 46

QUESTION 4.  Line 60. Rename microorganisms in italics.

Answer 4. Sorry again. Microorganisms have been renamed in italics                     according to the reviewer's indications in Line 60

 Results.

QUESTION 5. Line 105. Replace “de” by “the”.

Answer 5. This typo has been corrected as indicated by the reviewer, Line               110.

QUESTION 6 Lines 104-106. Rewrite the sentence. This statement is only         true for some microorganisms and natural products.

Answer 6. You are correct; there are some cases where this ratio has been determined to be greater than 1 (>1), which means it cannot be guaranteed that it will necessarily be less than 4. Therefore, the sentence has been modified as follows, Lines 109-111:

“The values of the ratio between the minimum bactericidal concentration (MBC) and the MIC of NE and TA indicated that the activity was bactericidal in most cases (MBC/MIC ≤ 4) for both compounds”.

QUESTION 7.   Standardise the units “mg/ml” or “mg/mL” throughout the                           manuscript.

Answer 7. The units indicated by the reviewer have been standardised throughout the document as mg/mL.

QUESTION 8. Table 1. Why are MBC values not well defined in cases of good antimicrobial activity?

Answer 8. You are correct. In the case of P. aerogenes, we were able to obtain a MIC value with TA, and we also had the MBC value, but it was mistakenly omitted from the table. This error has now been corrected in Table 1.

QUESTION 9 Tables 2, 3, 4. Include a footnote with the abbreviations mentioned in the table.

Answer 9. Following the reviewer's recommendations, a footnote has been included in each table (now Tables 1, 2, and 3) explaining the abbreviations as well as in Figures 1-4.

QUESTION 10. Figures. Improve the quality of the figures.

Answer 10. Following the reviewer's recommendations, the quality of the figures has been improved to 300 dpi.

QUESTION 11 Figure 1. Correct legend, replace NE by TA.

Answer 11. Of course, the erratum in figure 1 has been corrected.

QUESTION 12. Isobolograms. A more in-depth explanation of the figure and its relevance would be appreciated. 

Answer 12. Of course, the following text has been included at the bottom of                              the isobologram figures:

Figure 1. Isobolograms illustrate the interactions of Tannic Acid (TA) with the antibiotics (ABXs) where synergy was detected. The x-axis represents TA concentrations, while the y-axis represents antibiotic concentrations. The solid line, known as the 'addition line,' helps differentiate between additive effects—where points fall on or near this line—and synergistic effects, where concave isoboles are found below it. Additionally, there is a dashed line indicating the boundary of synergy. Points situated above or below this dashed line signify different degrees of synergistic interaction. Concentrations of ABXs and TA in (µg/mL)”.

Figure 2. Isobolograms illustrate the interactions of Nerol (NE) with the antibiotics (ABXs) where synergy was detected. The x-axis represents NE concentrations, while the y-axis represents ABX concentrations. The solid line, known as the 'addition line,' helps differentiate between additive effects—where points fall on or near this line—and synergistic effects, where concave isoboles are found below it. Additionally, there is a dashed line indicating the boundary of synergy. Points situated above or below this dashed line signify different degrees of synergistic interaction. Concentrations of ABXs and NE in (µg/mL)”.

The following text has been included in “Materials and methods. section 4.4.3, Lines 597-604.

“To draw an isobologram, the MIC of the NP is placed on the X-axis and the MIC of the ABX on the y-axis. The graph is plotted with the combinations obtained in the checkerboard tests that inhibit bacterial growth. It includes an 'addition line' (solid line) that helps differentiate between additive effects (where points fall above or near this line), synergistic effects (indicated by concave isoboles below the line), and antagonistic effects (shown by convex isoboles above the line). Additionally, there is a lower dotted line that marks the boundary of synergy. Points situated above or below this dotted line represent different degrees of synergistic interaction”

Finally, the discussion further elaborates on the information provided by the isobologram: Lines 231 –235:

“The synergies identified between natural products and antibiotics have been represented in isobolograms, which allow for a more intuitive visualization of the relationship between individual data points and a reference line (the addition line), making it easier to understand the degree of synergy for dose combinations that fall below this line”

Discussion

QUESTION 13. It is difficult to follow the discussion section. It is necessary to reorganise and synthesise, as well as to support the explanation with the corresponding tables or figures.

Answer 13. Of course, the discussion has been thoroughly revised. Introductory paragraphs have been added, explanations have been supported by referencing the relevant tables or figures, and responses have been provided to the points suggested below. Additionally, some paragraphs have been condensed. Changes can be seen in the tracked version of the manuscript.

QUESTION 14. Introduce the discussion of the results. Explain why you choose the natural products.

Answer 14. This justification is presented in the introduction (lines 56-76 first manuscript), where the antimicrobial properties of these natural products are emphasized, along with evidence suggesting their potential to produce synergies with antibiotics. Additionally, both products can be considered safe as they are already commercially available (lines 442-453) of the discussion.

However, you are correct that an introduction to this topic was needed in the discussion, which has now been added and expanded at the beginning (lines 183-194):

“In this study, we investigated the ability of two plant-derived natural products to synergize with widely used commercial ABXs, aiming to reduce the required ABX dosage while maintaining efficacy. TA and NE were selected based on their demonstrated anti-microbial activity against various Gram-positive and Gram-negative pathogens (F.A.Administration, 2024; Api et al., 2023; Friedman et al., 2002). Previous literature has also begun to suggest potential synergistic effects of these compounds with one or a few ABXs (Stumpf et al., 2020). The use of these products offers a range of advantages: NE is abundant in essential oils from widely cultivated plants (Verma et al., 2020) while TA is prevalent in the bark of trees like oak, chestnut mahogany (Caprarulo et al., 2021). Therefore, their extraction from these inexpensive plant materials ensures high availability. Moreover, the essential oil industry enables cost-effective NE production, often as a byproduct of other compounds. Similarly, byproducts from the wine and wood industries can be used to extract TA, promoting sustainable production (Conidi et al., 2022)”.

QUESTION 15. Justify why you use only one strain of each microorganism.

Answer 15. The study was conducted with 14 different microorganisms, including both Gram-positive and Gram-negative bacteria, as well as a unicellular fungus. These organisms were chosen because they are pathogens with high incidence in human health, animal health, or food safety, as mentioned in the last paragraph of the introduction (Lines 82-91, first manuscript). The strains used are reference strains widely employed in research and recommended as representative by the laboratories from which they were obtained.

Studying the antimicrobial effects of the two products on this set of bacteria already represents a significant experimental workload. Additionally, all possible synergies between the two natural products and eight representative commercial antibiotics, chosen for their high consumption, were tested. This resulted in a total of 48 combinations, for which the most effective synergies were further analyzed by tracking their growth kinetics.

We respectfully believe that incorporating different strains of the same bacteria into the study would have unnecessarily complicated the experimental workload. In fact, upon reviewing the literature, the few microorganisms for which NE or TA have been applied show MIC values that are quite similar for the same bacteria, even when different strains are used (Lines 215-224, first manuscript). While it is true that some differences in MIC were found, it is difficult to determine whether these are due to the strain, the technique used, the solvent, or the culture medium (Lines 223-224), first manuscript.

One aspect we do have in mind for future studies is the use of resistant strains obtained from human or veterinary clinics, as this could provide additional valuable information. However, before exploring these variants, we believe it was necessary to conduct this study with standard reference strains, which allows for comparison of results with those from the literature, such as antimicrobial activity and synergies, with other natural products

Taking your comment into account, however, a clarification has been added to the discussion (lines 195-201):

“The antimicrobial capacity of both products was tested against a comprehensive and representative panel of 14 pathogenic bacteria to determine the MICs. This allowed us to conduct a synergy study with 8 ABXs, resulting in a total of 48 combinations. The synergies (4 for NE and 6 for TA) were further analyzed by monitoring their growth kinetics. A reference bacterial strain was selected for each bacterium studied, enabling comparison with results from the literature, such as those involving other natural products or combinations with different ABXs.”.

QUESTION 16. Explain why natural products do not contribute to antimicrobial resistance.

Answer 16. Thank you for the suggestion. Explanatory paragraphs on this point have been included in the discussion: lines (422-441):

“In addition to the previously mentioned advantages in their production, TA and NE likely have a lower potential for inducing resistance compared to commercial ABXs due to several factors (Sharma et al., 2024). First, natural compounds often have more complex and diverse chemical structures than synthetic ABXs, making it harder for bacteria to develop effective resistance mechanisms as they would need to adapt to multiple sites or modes of action, which is more challenging. Additionally, while synthetic ABXs typically target a single cellular process, natural products like TA and NE act on multiple fronts as we have seen. This multifaceted approach further reduces the likelihood of resistance, as bacteria would need to simultaneously mutate in several areas. Moreover, NE and TA can disrupt bacterial membranes, affecting resistance mechanisms associated with these membranes, such as efflux pumps. Another key point is that natural compounds have been in contact with microorganisms for millennia, possibly leading to an evolutionary balance where bacteria are less prone to develop resistance. In contrast, synthetic ABXs are more recent and often used in large quantities, which can exert intense selective pressure, quickly fostering the development of resistant strains. When natural products are used in synergy with ABXs, the required ABX doses can be significantly reduced, as seen in this study, decreasing the chances of resistance development. Finally, the simultaneous action of natural products and ABXs on different cellular targets makes it more difficult for bacteria to develop resistance strategies against both, a benefit that is less common in synthetic ABXs typically used as monotherapies”

QUESTION 17. Is the concentration range of the natural product-antibiotic combination appropriate for clinical use? Include potential applications of the natural compounds in combination with antibiotics.

Answer 17. Thanks for the suggestion. A paragraph discussing the potential applications of the synergies has been included in the discussion, as suggested, lines 454-480):

In general, natural products require higher doses compared to ABXs to achieve antimicrobial activity. However, the advantage of synergistic combinations is that they can reduce not only the concentration of ABXs but also the amount of the natural product needed, as demonstrated by our results.

Typically, these synergies are considered for use in topical formulations such as lotions, ointments, gels, or creams for skin infections, wounds, and ulcers (Sharma et al., 2024). They could also be applied in mouthwashes and oral rinses (Gan et al., 2023). Oral solutions like Cesinex®  (Ren et al., 2012) are already marketed for gastrointestinal conditions, with tannic acid as one of their components.

Additionally, there is potential for these combinations in veterinary feed to treat animal diseases, which is relevant given EU Regulation 2019/6 (Union, 2019), which emphasizes reducing ABX use in livestock to mitigate environmental impact (Pino-Otin et al., 2024) and microbial resistance (Teuber, 2001). Furthermore, these synergies could be valuable as disinfectants to inhibit microbial biofilm formation on stainless steel surfaces (Myszka et al., 2016).

The effective doses found in the synergies between natural products and ABXs are very low, ranging from 23.44 to 81.25 μg/mL for TA and from 125 to 250 μg/mL for NE. The ABXs in synergy further reduce their concentrations significantly, with ranges between 0.78 and 31.25 μg/mL. Topical ABXs available on the market, for example, are used at concentrations several orders of magnitude higher than the synergistic concentrations obtained in this study. Most of the available ABX ointments (such as those with mupirocin or sodium fusidate) and antifungal ointments (like ketoconazole) contain a concentration of 20 mg/g. Other formulations contain even more, such as oxytetracycline or clindamycin ointments, which typically have 30 mg/g of the ABX. Commercially available ABX eye drops have similar concentrations, often 3 mg/mL (or 3000 µg/mL). Some formulations of gentamicin even reach up to 5000 µg/mL of the ABX. Therefore, it is pharmacologically realistic to prepare topical, veterinary, or even oral formulations containing these ABXs and natural products at the effective concentrations found in the synergies of our study.

(...)

lines 489-495

“However, before developing clinical formulations, it is crucial to assess the potential toxicity of the combination, as well as its bioavailability, through clinical trials (Langeveld et al., 2014), since the physicochemical and pharmacological properties of these combinations may differ from those of the individual products (Cushnie et al., 2014) . Formulations must also take into account organoleptic properties and stability. Finally, within a One Health framework, it is important to evaluate whether these synergies are more environmentally friendly compared to using higher doses of ABXs alone”.

Materials and methods

QUESTION 18. As the Materials and methods section appears at the end of the manuscript, all abbreviations should be defined in the Results section.

Answer 18. The designation of antibiotic by ABX has been unified throughout the text.

QUESTION 19. Line 415. Replace “Table 1” by “Table 5”.

Answer 19. The former Table 5 is now Table 4, since the table of antimicrobial capacities of the antibiotics (previously Table 2) has been moved to the supplementary material (Support Information 1). Therefore, 'Table 1' has been replaced with 'Table 4”.

QUESTION 20. Table 5. Include the range of concentrations tested.

Answer 20. Following the reviewer's recommendation, a column has been included in this table (now table 4) with the range of concentrations used for each antibiotic.

QUESTION 21. Line 465. Which McFarland standard was used?

Answer 21. The authors indicate the standard used to perform in Mcfarland in the following paragraph on lines 541-543

“Using a BioTekTM Synergy H1 hybrid multimode microplate reader (625 nm), bacterial cultures were previously adjusted to the McFarland standard (CLSI, 2018) [1] to achieve a standard initial bacterial concentration per well of roughly 2.5 x 10^5 CFU/mL.”

QUESTION 22. Line 478. Replace “tasted” by “tested”.

Answer 22. Of course, ‘tasted’ has been replaced by ‘tested’ line 557

QUESTION 23. Section 4.2. Include a table with the details of the tested strains (collection number, media and temperature for the antimicrobial experiments)

Answer 23. Of course, a table with this information has been added as support Information 2

Support Information 2. Microorganisms reference and culture conditions according to American Type Culture Collection (ATCC) datasheets for each microorganism.

Microorganism culture conditions

Microorganism

Reference

GRAM

Temperature (ºC)

Time (h)

Agar/Broth

Atmosphere

Bacillus subtilis

ATCC  6633

positive

30

24

BHI

Aerobic

Staphylococcus aureus

ATCC  9144

37

TS

Enterococcus faecalis

ATCC 19433

BHI

Streptococcus agalactiae

ATCC 12386

Escherichia coli

ATCC 25922

negative

TS

NU

Klebsiella pneumoniae

C6

Serratia marcescens subsp.marcescens

ATCC 13880

26

24-48

Proteus mirabilis

ATCC 35659

37

24

TS

Pseudomonas aeruginosa

ATCC 27853

Klebsiella aerogenes

ATCC 13048

30

NU

Acinetobacter baumannii

ATCC 19606

37

Pasteurella aerogenes

ATCC 27883

BHI

Salmonella entérica

ATCC 13311

NU

Acinetobacter baumannii

ATCC 19606

Candida albicans

ATCC 10231

25-30

SDB

TS -Trypticase Soy Agar/Broth, NU - Nutrient agar or nutrient broth, BHI -  Brain Heart Infusion Agar/Broth, SDB - Sabouraud Dextrose Broth

QUESTION 24. Sections 4.3 and 4.4. Reduce these sections by eliminating unnecessary details for the experimental set-up such as microplate configuration.

ANSWER 24.  Following the reviewer's instructions, the details of the microplate configuration in sections 4.3 and 4.4 have been removed. The following paragraphs have been deleted from the text.

Section 4.3

“Each well contained 100µL of the correct media for each bacteria, according ATCC product sheet instructions. The first column of each microplate was then filled with 100µL of the NP or ABX stock solution, and after applying serial two-fold dilution from columns 1 to 10, the volume obtained for each well was 100µL. In columns 11 and 12, respectively, each experiment included a positive control measuring bacterial growth and a negative control assessing sterile conditions. Finally, 10 µL of bacterial inoculum was added in each well (excluding column 12).”

Section 4.4

The plates were next inoculated with 10µL of bacterial suspension, previously prepared in accordance with Section 4.3 and adjusted to the McFarland standard. As indicated in Section 4.3, the plates were incubated for 24 hours at the adequate temperature specified for each strain before absorbance measurement (625 nm) to assess bacterial growth.

New references:

Abreu AC, McBain AJ, Simoes M. Plants as sources of new antimicrobials and resistance-modifying agents. Natural Product Reports 2012; 29: 1007-1021.

Administration FaD. Food additives permitted for direct addiction to food for human consumption. In: Home. F, editor. 21, 2024.

Ahmad MF, Alsayegh AA, Ahmad FA, Akhtar MS, Alavudeen SS, Bantun F, et al. <i>Ganoderma</i><i> lucidum:</i> Insight into antimicrobial and antioxidant properties with development of secondary metabolites. Heliyon 2024; 10.

Api A, Belsito D, Botelho D, Bruze M, Burton G, Cancellieri M, et al. RIFM fragrance ingredient safety assessment, nerol, CAS Registry Number 106-25-2. FOOD AND CHEMICAL TOXICOLOGY 2023; 179.

Assane IM, Santos NA, de Sousa EL, Brasil M, Cilli EM, Pilarski F. Cytotoxicity and antimicrobial activity of synthetic peptides alone or in combination with conventional antimicrobials against fish pathogenic bacteria. Journal of Applied Microbiology 2021; 131: 1762-1774.

Basile A, Giordano S, López-Sáez JA, Cobianchi RC. Antibacterial activity of pure flavonoids isolated from mosses. Phytochemistry 1999; 52: 1479-1482.

Betoni JEC, Passarelli Mantovani R, Nunes Barbosa L, Di Stasi LC, Fernandes Junior A. Synergism between plant extract and antimicrobial drugs used on Staphylococcus aureus diseases. Memorias Do Instituto Oswaldo Cruz 2006; 101: 387-390.

Caprarulo V, Giromini C, Rossi L. Review: Chestnut and quebracho tannins in pig nutrition: the effects on performance and intestinal health. Animal 2021; 15.

Chan BCL, Ip M, Gong H, Lui SL, See RH, Jolivalt C, et al. Synergistic effects of diosmetin with erythromycin against ABC transporter over-expressed methicillin-resistant <i>Staphylococcus aureus</i> (MRSA) RN4220/pUL5054 and inhibition of MRSA pyruvate kinase. Phytomedicine 2013; 20: 611-614.

Chovanova R, Mikulasova M, Vaverkova S. In Vitro Antibacterial and Antibiotic Resistance Modifying Effect of Bioactive Plant Extracts on Methicillin-Resistant Staphylococcus epidermidis. International journal of microbiology 2013; 2013: 760969-760969.

Chung KT, Wong TY, Wei CI, Huang YW, Lin Y. Tannins and human health: A review. Critical Reviews in Food Science and Nutrition 1998; 38: 421-464.

Conidi C, Donato L, Algieri C, Cassano A. Valorization of chestnut processing by-products: A membrane-assisted green strategy for purifying valuable compounds from shells. JOURNAL OF CLEANER PRODUCTION 2022; 378.

Coutinho HDM, Costa JGM, Falcao-Silva VS, Siqueira JP, Jr., Lima EO. Effect of Momordica charantia L. in the resistance to aminoglycosides in methicilin-resistant Staphylococcus aureus. Comparative Immunology Microbiology and Infectious Diseases 2010; 33: 467-471.

Cushnie TPT, Cushnie B, Lamb AJ. Alkaloids: An overview of their antibacterial, antibiotic-enhancing and antivirulence activities. International Journal of Antimicrobial Agents 2014; 44: 377-386.

Daud NM, Putra NR, Jamaludin R, Norodin NSM, Sarkawi NS, Hamzah MHS, et al. Valorisation of plant seed as natural bioactive compounds by various extraction methods: A review. Trends in Food Science & Technology 2022; 119: 201-214.

de Oliveira MS, da Cruz JN, da Costa WA, Silva SG, Brito MdP, Fernandes de Menezes SA, et al. Chemical Composition, Antimicrobial Properties of<i>Siparuna guianensis</i>Essential Oil and a Molecular Docking and Dynamics Molecular Study of its Major Chemical Constituent. Molecules 2020; 25.

Drioiche A, Baammi S, Zibouh K, Al Kamaly O, Alnakhli AM, Remok F, et al. A Study of the Synergistic Effects of Essential Oils from <i>Origanum compactum</i> and <i>Origanum elongatum</i> with Commercial Antibiotics against Highly Prioritized Multidrug-Resistant Bacteria for the World Health Organization. Metabolites 2024; 14.

Du Y, Huo Y, Yang Q, Han Z, Hou L, Cui B, et al. Ultrasmall iron-gallic acid coordination polymer nanodots with antioxidative neuroprotection for PET/MR imaging-guided ischemia stroke therapy. Exploration 2023; 3.

Ferrando N, Pino-Otin MR, Ballestero D, Lorca G, Terrado EM, Langa E. Enhancing Commercial Antibiotics with Trans-Cinnamaldehyde in Gram-Positive and Gram-Negative Bacteria: An In Vitro Approach. Plants-Basel 2024; 13.

Friedman M, Henika PR, Mandrell RE. Bactericidal activities of plant essential oils and some of their isolated constituents against <i>Campylobacter jejuni</i>, <i>Escherichia coli</i>, <i>Listeria monocytogenes</i>, and <i>Salmonella enterica</i>. Journal of Food Protection 2002; 65: 1545-1560.

Gan C, Langa E, Valenzuela A, Ballestero D, Rosa Pino-Otin M. Synergistic Activity of Thymol with Commercial Antibiotics against Critical and High WHO Priority Pathogenic Bacteria. Plants-Basel 2023; 12.

Gulcin I, Huyut Z, Elmastas M, Aboul-Enein HY. Radical scavenging and antioxidant activity of tannic acid. Arabian Journal of Chemistry 2010; 3: 43-53.

Guo Z, Xie W, Lu J, Guo X, Xu J, Xu W, et al. Tannic acid-based metal phenolic networks for bio-applications: a review. JOURNAL OF MATERIALS CHEMISTRY B 2021; 9: 4098-4110.

Iacobellis NS, Lo Cantore P, Capasso F, Senatore F. Antibacterial activity of <i>Cuminum cyminum</i> L. and <i>Carum carvi</i> L. essential oils. Journal of Agricultural and Food Chemistry 2005; 53: 57-61.

Jha AK, Sit N. Extraction of bioactive compounds from plant materials using combination of various novel methods: A review. Trends in Food Science & Technology 2022; 119: 579-591.

Jing W, Xiaolan C, Yu C, Feng Q, Haifeng Y. Pharmacological effects and mechanisms of tannic acid. BIOMEDICINE & PHARMACOTHERAPY 2022; 154.

Kalemba D, Kunicka A. Antibacterial and antifungal properties of essential oils. Current Medicinal Chemistry 2003; 10: 813-829.

Kasthuri T, Swetha TK, Bhaskar JP, Pandian SK. Rapid-killing efficacy substantiates the antiseptic property of the synergistic combination of carvacrol and nerol against nosocomial pathogens. Archives of Microbiology 2022; 204.

Keck JM, Viteri A, Schultz J, Fong R, Whitman C, Poush M, et al. New Agents Are Coming, and So Is the Resistance. Antibiotics-Basel 2024; 13.

King A, Young G. Characteristics and occurrence of phenolic phytochemicals. Journal of the American Dietetic Association 1999; 99: 213-218.

Kotan R, Kordali S, Cakir A. Screening of antibacterial activities of twenty-one oxygenated monoterpenes. Zeitschrift Fur Naturforschung Section C-a Journal of Biosciences 2007; 62: 507-513.

Kumar S, Mukherjee A, Dutta J. Chitosan based nanocomposite films and coatings: Emerging antimicrobial food packaging alternatives. Trends in Food Science & Technology 2020; 97: 196-209.

Langeveld WT, Veldhuizen EJA, Burt SA. Synergy between essential oil components and antibiotics: a review. Critical Reviews in Microbiology 2014; 40: 76-94.

Lapczynski A, Foxenberg RJ, Bhatia SP, Letizia CS, Api AM. Fragrance material review on nerol. Food and Chemical Toxicology 2008; 46: S241-S244.

Lederer S, Dijkstra TMH, Heskes T. Additive Dose Response Models: Defining Synergy. Frontiers in Pharmacology 2019; 10.

Leonard CM, Virijevic S, Regnier T, Combrinck S. Bioactivity of selected essential oils and some components on <i>Listeria monocytogenes</i> biofilms. South African Journal of Botany 2010; 76: 676-680.

Li Y, Duan Y, Li Y, Gu Y, Zhou L, Xiao Z, et al. Cascade loop of ferroptosis induction and immunotherapy based on metal-phenolic networks for combined therapy of colorectal cancer. Exploration 2024.

Lopes GKB, Schulman HM, Hermes-Lima M. Polyphenol tannic acid inhibits hydroxyl radical formation from Fenton reaction by complexing ferrous ions. Biochimica Et Biophysica Acta-General Subjects 1999; 1472: 142-152.

Lupia C, Castagna F, Bava R, Naturale MD, Zicarelli L, Marrelli M, et al. Use of Essential Oils to Counteract the Phenomena of Antimicrobial Resistance in Livestock Species. Antibiotics-Basel 2024; 13.

Macia MD, Rojo-Molinero E, Oliver A. Antimicrobial susceptibility testing in biofilm-growing bacteria. Clinical Microbiology and Infection 2014; 20: 981-990.

Masyita A, Sari RM, Astuti AD, Yasir B, Rumata NR, Bin Emran T, et al. Terpenes and terpenoids as main bioactive compounds of essential oils, their roles in human health and potential application as natural food preservatives. Food Chemistry-X 2022; 13.

McEwen SA, Collignon PJ. Antimicrobial Resistance: a One Health Perspective. In: Schwarz S, Cavaco LM, Shen J, editors. Antimicrobial Resistance in Bacteria from Livestock and Companion Animals, 2018, pp. 521-547.

Myszka K, Schmidt MT, Majcher M, Juzwa W, Olkowicz M, Czaczyk K. Inhibition of <i>quorum sensing</i>-related biofilm of <i>Pseudomonas fluorescens</i> KM121 by <i>Thymus vulgare</i> essential oil and its major bioactive compounds. International Biodeterioration & Biodegradation 2016; 114: 252-259.

Noel DJ, Keevil CW, Wilks SA. Synergism versus Additivity: Defining the Interactions between Common Disinfectants. Mbio 2021; 12.

Olson ME, Ceri H, Morck DW, Buret AG, Read RR. Biofilm bacteria: formation and comparative susceptibility to antibiotics. Canadian Journal of Veterinary Research-Revue Canadienne De Recherche Veterinaire 2002; 66: 86-92.

Orlowski P, Zmigrodzka M, Tomaszewska E, Ranoszek-Soliwoda K, Czupryn M, Antos-Bielska M, et al. Tannic acid-modified silver nanoparticles for wound healing: the importance of size. International Journal of Nanomedicine 2018; 13: 991-1007.

Othman L, Sleiman A, Abdel-Massih RM. Antimicrobial Activity of Polyphenols and Alkaloids in Middle Eastern Plants. Frontiers in Microbiology 2019; 10.

Park JH, Choi S, Moon HC, Seo H, Kim JY, Hong S-P, et al. Antimicrobial spray nanocoating of supramolecular Fe(III)-tannic acid metal-organic coordination complex: applications to shoe insoles and fruits. Scientific Reports 2017; 7.

Pino-Otin MR, Valenzuela A, Gan C, Lorca G, Ferrando N, Langa E, et al. Ecotoxicity of five veterinary antibiotics on indicator organisms and water and soil communities. Ecotoxicology and Environmental Safety 2024; 274.

Ren A, Zhang W, Thomas HG, Barish A, Berry S, Kiel JS, et al. A Tannic Acid-Based Medical Food, Cesinex<SUP>A®</SUP>, Exhibits Broad-Spectrum Antidiarrheal Properties: A Mechanistic and Clinical Study. Digestive Diseases and Sciences 2012; 57: 99-108.

Sharma A, Anurag J, Kaur J, Kesharwani A, Parihar VK. Antimicrobial Potential of Polyphenols: An Update on Alternative for Combating Antimicrobial Resistance. Medicinal Chemistry 2024; 20: 576-596.

Si H, Hu J, Liu Z, Zeng Z-l. Antibacterial effect of oregano essential oil alone and in combination with antibiotics against extended-spectrum β-lactamase-producing <i>Escherichia coli</i>. Fems Immunology and Medical Microbiology 2008; 53: 190-194.

Sibanda T, Okoh AI. The challenges of overcoming antibiotic resistance: Plant extracts as potential sources of antimicrobial and resistance modifying agents. African Journal of Biotechnology 2007; 6: 2886-2896.

Stumpf S, Hostnik G, Primozic M, Leitgeb M, Salminen J-P, Bren U. The Effect of Growth Medium Strength on Minimum Inhibitory Concentrations of Tannins and Tannin Extracts against<i>E. coli</i>. Molecules 2020; 25.

Taylor PW. Alternative natural sources for a new generation of antibacterial agents. International Journal of Antimicrobial Agents 2013; 42: 195-201.

Teuber M. Veterinary use and antibiotic resistance. Current Opinion in Microbiology 2001; 4: 493-499.

Turek C, Stintzing FC. Stability of Essential Oils: A Review. Comprehensive Reviews in Food Science and Food Safety 2013; 12: 40-53.

Union E. Regulation (EU) 2019/6 of the European Parliament and of the Council of 11 December 2018 on veterinary medicinal products and repealing Directive 2001/82/EC, 2019.

van Vuuren SF, Suliman S, Viljoen AM. The antimicrobial activity of four commercial essential oils in combination with conventional antimicrobials. Letters in Applied Microbiology 2009; 48: 440-446.

Vaou N, Stavropoulou E, Voidarou C, Tsakris Z, Rozos G, Tsigalou C, et al. Interactions between Medical Plant-Derived Bioactive Compounds: Focus on Antimicrobial Combination Effects. Antibiotics-Basel 2022; 11.

Vaou N, Stavropoulou E, Voidarou C, Tsigalou C, Bezirtzoglou E. Towards Advances in Medicinal Plant Antimicrobial Activity: A Review Study on Challenges and Future Perspectives. Microorganisms 2021; 9.

Verma A, Srivastava R, Sonar P, Yadav R. Traditional, phytochemical, and biological aspects of <i>Rosa alba</i> L.: a systematic review. FUTURE JOURNAL OF PHARMACEUTICAL SCIENCES 2020; 6.

Wagner H, Ulrich-Merzenich G. Synergy research: Approaching a new generation of phytopharmaceuticals. Phytomedicine 2009; 16: 97-110.

Wang Z, Yang K, Chen L, Yan R, Qu S, Li Y-x, et al. Activities of Nerol, a natural plant active ingredient, against <i>Candida albicans</i> in vitro and in vivo. Applied Microbiology and Biotechnology 2020; 104: 5039-5052.

Wu LT, Chu CC, Chung JG, Chen CH, Hsu LS, Liu JK, et al. Effects of tannic acid and its related compounds on food mutagens or hydrogen peroxide-induced DNA strands breaks in human lymphocytes. Mutation Research-Fundamental and Molecular Mechanisms of Mutagenesis 2004; 556: 75-82.

Yang S-K, Yusoff K, Mai C-W, Lim W-M, Yap W-S, Lim S-HE, et al. Additivity vs. Synergism: Investigation of the Additive Interaction of Cinnamon Bark Oil and Meropenem in Combinatory Therapy. Molecules 2017; 22.

Yap PSX, Lim SHE, Hu CP, Yiap BC. Combination of essential oils and antibiotics reduce antibiotic resistance in plasmid-conferred multidrug resistant bacteria. Phytomedicine 2013; 20: 710-713.

Yeshi K, Crayn D, Ritmejeryte E, Wangchuk P. Plant Secondary Metabolites Produced in Response to Abiotic Stresses Has Potential Application in Pharmaceutical Product Development. Molecules 2022; 27.

Zych S, Adaszynska-Skwirzynska M, Szewczuk MA, Szczerbinska D. Interaction between Enrofloxacin and Three Essential Oils (Cinnamon Bark, Clove Bud and Lavender Flower)-A Study on Multidrug-Resistant <i>Escherichia coli</i> Strains Isolated from 1-Day-Old Broiler Chickens. International Journal of Molecular Sciences 2024; 25.

Round 2

Reviewer 3 Report

Comments and Suggestions for Authors

No additional comments